# Phosphorylation-dependent pseudokinase domain dimerization drives full-length MLKL oligomerization

Yanxiang Meng [1,2,10], Sarah E. Garnish[1,2,10], Katherine A. Davies [1,2], Katrina A. Black [1,2], Andrew P. Leis[1,2], Christopher R. Horne [1,2], Joanne M. Hildebrand [1,2], Hanadi Hoblos[1,2], Cheree Fitzgibbon[1,2], Samuel N. Young[1], Toby Dite[1,2], Laura F. Dagley [1,2], Aarya Venkat[3], Natarajan Kannan [3,4], Akiko Koide[5,6], Shohei Koide [5,7], Alisa Glukhova [1,2,8,9], Peter E. Czabotar [1,2] ✉ & James M. Murphy [1,2,8] ✉

The necroptosis pathway is a lytic, pro-inflammatory mode of cell death that is widely implicated in human disease, including renal, pulmonary, gut and skin inflammatory pathologies. The precise mechanism of the terminal steps in the pathway, where the RIPK3 kinase phosphorylates and triggers a conformation change and oligomerization of the terminal pathway effector, MLKL, are only emerging. Here, we structurally identify RIPK3-mediated phosphorylation of the human MLKL activation loop as a cue for MLKL pseudokinase domain dimerization. MLKL pseudokinase domain dimerization subsequently drives formation of elongated homotetramers. Negative stain electron microscopy and modelling support nucleation of the MLKL tetramer assembly by a central coiled coil formed by the extended, ~80 Å brace helix that connects the pseudokinase and executioner four-helix bundle domains. Mutational data assert MLKL tetramerization as an essential prerequisite step to enable the release and reorganization of four-helix bundle domains for membrane permeabilization and cell death.

Cell death via the necroptosis pathway is thought to have arisen as an altruistic means of countering pathogens[1–8]. More recently, it has become evident that dysregulated necroptotic signalling may contribute to, or promote, many inflammatory diseases[9–11], including those of the gut[12–14], skin[15,16], kidney[17–19] and lung[20], raising interest in therapeutic targeting of the pathway.

Necroptosis can be initiated upon ligation of death receptors, such as Tumour Necrosis Factor (TNF) receptor 1 (TNFR1) by TNF[21,22],

or the Toll-like receptor (TLR)−3/4 or ZBP1 pathogen receptors by pathogen-associated molecular patterns (PAMPs)[23,24]. Downstream, receptor-interacting serine/threonine protein kinase (RIPK)−1 scaffolds assembly of a high molecular weight signalling platform with RIPK3 kinase termed the necrosome[25], from which necroptosis is launched[26]. Formation of the necrosome arises in cellular contexts where cIAP (cellular inhibitors of apoptosis)-mediated ubiquitylation[27] and Caspase-8's proteolytic activity[28] are suppressed to prevent RIPK1

[1]Walter and Eliza Hall Institute of Medical Research, 1G Royal Parade, Parkville, VIC 3052, Australia. [2]Department of Medical Biology, University of Melbourne, Parkville, VIC 3052, Australia. [3]Institute of Bioinformatics, University of Georgia, Athens, GA 30602, USA. [4]Department of Biochemistry and Molecular Biology, University of Georgia, Athens, GA 30602, USA. [5]Perlmutter Cancer Center, New York University Langone Health, New York, NY 10016, USA. [6]Department of Medicine, New York University School of Medicine, New York, NY 10016, USA. [7]Department of Biochemistry and Molecular Pharmacology, New York University School of Medicine, New York, NY 10016, USA. [8]Drug Discovery Biology, Monash Institute of Pharmaceutical Sciences, Monash University, Parkville, VIC 3052, Australia. [9]Department of Biochemistry and Pharmacology, University of Melbourne, Parkville, VIC 3052, Australia. [10]These authors contributed equally: Yanxiang Meng, Sarah E. Garnish. ✉e-mail: czabotar@wehi.edu.au; jamesm@wehi.edu.au

engagement in NF-κB and apoptosis signalling, respectively. A cytoplasmic complex of RIPK3 and the terminal executioner protein, MLKL (mixed lineage kinase domain-like)[29,30], is subsequently recruited to the necrosome, where RIPK3-mediated phosphorylation triggers MLKL activation[30–32]. Phosphorylation of the activation loop within the MLKL pseudokinase domain serves two key functions in MLKL activation: prompting conformational interconversion; and inducing release of MLKL from the necrosome[33]. MLKL appears to assemble into oligomers at the necrosome, and these oligomers are trafficked to the plasma membrane with the assistance of chaperones and the actin, Golgi and microtubule trafficking machinery[34]. At the membrane, MLKL accumulates into hotspots that, once a critical threshold is exceeded, permeabilize the lipid bilayer[34], which leads to cell swelling and, eventually, lysis and expulsion of the cellular contents into the extracellular milieu where they function as immunogenic alarmins[35].

To date, the only full-length structure of a necroptosis pathway protein to be reported is of the mouse MLKL monomer[32]. However, because of substantial differences between the mouse and human MLKL activation mechanisms[29,36–41], mouse MLKL is an unsuitable model for understanding human MLKL regulation. Currently, the only structural information reported for human MLKL is of the isolated component domains[42,43], as is the case with RIPK1 and RIPK3[29,44–46]. Thus, the precise arrangement of MLKL's component domains within the full-length protein, and how human MLKL assembles into oligomers to mediate cell death is incompletely understood. Additionally, the stoichiometry of the oligomers formed by recombinant MLKL has been debated over the past decade, and currently the stoichiometry of cytoplasmic MLKL clusters and membrane-associated hotspots has not been ascertained. Biochemical data implicate the brace helices that connect the N-terminal executioner, four-helix bundle (4HB) domain and the C-terminal regulatory, pseudokinase domain in mediating MLKL oligomerization[37,47], although the organization of the component domains within oligomers has not yet been visualized. Furthermore, it has remained mysterious why full-length recombinant human MLKL exhibits vastly greater potency in permeabilizing membrane bilayers in vitro than the isolated recombinant 4HB domain of human MLKL[41], despite the latter's function as the membrane disrupting domain[40,43,48–50].

Here we present the crystal structure of the phosphorylated human MLKL pseudokinase domain, which we validated biophysically to form a head-to-tail, back-to-back dimer structure in solution. A pair of MLKL pseudokinase dimers reside at the C-terminal end of an elongated, full-length MLKL tetramer structure that we predicted using AlphaFold modelling and validated by negative stain electron microscopy. The pseudokinase domain connects to the brace region, which forms a single long helix spanning ~80 Å, rather than the two shorter helices observed in the full-length mouse MLKL structure[32]. The brace helices from the four subunits pack against one another to form an extended coiled coil, which connect to an assembly of four 4HB domains via flexible linker sequences. Our data support the idea that RIPK3-mediated phosphorylation drives the MLKL pseudokinase domain dimerization and release of the brace helices into an extended single helix that nucleates MLKL assembly into pro-necroptotic tetramers. Mutagenesis data support the critical role of 4HB domain unlatching from an autoinhibitory interaction with the brace helices within these tetramers to enable the 4HB domains to rearrange into a previously-unreported tetrameric bundle that can potently permeabilize membranes.

## Results

### RIPK3-mediated phosphorylation drives human MLKL pseudokinase domain dimerization

To investigate the mechanisms by which RIPK3-mediated phosphorylation of the activation loop residues, T357 and S358, within the human MLKL pseudokinase domain drives necroptosis, we prepared recombinant phosphorylated human MLKL (p-MLKL) pseudokinase domain by co-expressing MLKL with human RIPK3 kinase domain in Sf21 insect cells (Supplementary Fig. 1a, b). Intact protein mass spectrometry confirmed that the purified p-MLKL pseudokinase domain was phosphorylated, with between one to three phosphosites per protomer (Supplementary Fig. 1b). These phosphorylation events occurred at T355, T357, S358 and S360 within the activation loop (Supplementary Fig. 1c–e), with no other phosphosites detected by mass spectrometry elsewhere in the MLKL pseudokinase domain or in the absence of RIPK3-mediated phosphorylation. Unexpectedly, p-MLKL pseudokinase domain eluted off size-exclusion chromatography at a retention volume consistent with a dimer, in contrast to the unphosphorylated, monomeric MLKL pseudokinase domain[42] (Supplementary Fig. 1a). Dephosphorylation of p-MLKL pseudokinase domain with λ Phosphatase led to elution at a retention time comparable to that of the monomeric pseudokinase domain (Supplementary Fig. 1f), indicating dimerization is reliant on MLKL pseudokinase domain phosphorylation and is reversible.

Previously, we proposed that the MLKL pseudokinase domain undergoes a conformational change to adopt a closed, active kinase-like conformation upon RIPK3-mediated phosphorylation[29,33]. To formally examine whether the p-MLKL pseudokinase domain adopts a closed conformation, we determined the crystal structure of p-MLKL pseudokinase domain to 2.3 Å resolution (Supplementary Table 1 and Fig. 1a, left), with one copy of p-MLKL in the asymmetric unit in the C2 2 2₁ space group. The p-MLKL pseudokinase domain forms a large interface with a symmetry mate via the pseudokinase domain hinge region (1118.6 Å²; calculated using PISA[51]), forming a head-to-tail, back-to-back dimer in crystallo (Fig. 1b). This dimer model closely resembles the pseudokinase domain dimer proposed based on the phosphomimetic T357D/S358E mutant MLKL structure (PDB: 6LK5)[52]. To validate this model, we examined the structure of this recombinant p-MLKL pseudokinase domain in solution using small-angle X-ray scattering (SAXS) coupled with inline size-exclusion chromatography. The radius of gyration ($R_g$) and maximum dimension ($D_{max}$) are consistent with the size expected of a pseudokinase domain dimer, and they are considerably larger than the values previously reported for wild-type, unliganded, unphosphorylated (apo) MLKL pseudokinase domain[42] (Table 1). The predicted scattering profile calculated from the p-MLKL pseudokinase domain dimer structure was in close agreement with the experimental scattering profile (Fig. 1c), indicating that this dimer model formed with a crystallographic symmetry mate reflects the oligomeric state and organization of p-MLKL pseudokinase domain in solution.

The p-MLKL pseudokinase domain adopts a closed conformation (PDB: 8SLZ; Supplementary Fig. 2a, exhibiting classical active kinase features including a salt bridge between the β3 strand Lys (VAIK230) and the αC helix Glu (E250) and aligned regulatory (R)-spine residues[53] (Fig. 1a). Prior crystal structures of the human MLKL pseudokinase domain in this active-like conformation[40,42,52,54–56] (Table 2) show strong concordance with the p-MLKL structure reported herein, supporting the idea that the closed conformer represents the activated form of MLKL. The closed form differs from the open conformation observed in the MLKL:RIPK3 complex[29] (PDB: 7MON; Supplementary Fig. 2b), where the activation loop instead forms a helix that buttresses against, and displaces, the αC helix from a position typical of canonical active kinase structures. In the crystal structure of human p-MLKL, the activation loop (residues 355–368) was highly flexible and could not be modelled due to a lack of electron density. The flexibility of the activation loop is consistent with previous reports of human MLKL pseudokinase domains (PDB: 4MWI, 4M67, 6LK6, 6LK5, 6BWK, 5KO1, 5KNJ, 6O5Z, 7JW7)[40,42,52,54–56] (Table 2) and is commonly observed in protein kinase domains[57]. As a result, phosphorylation of T357/S358 of MLKL could not be directly observed in our structure. Unlike in the inactive RIPK3:MLKL complex structure[29] where the activation loop harbouring the RIPK3 substrates, T357 and S358, forms a short α-helix, T357

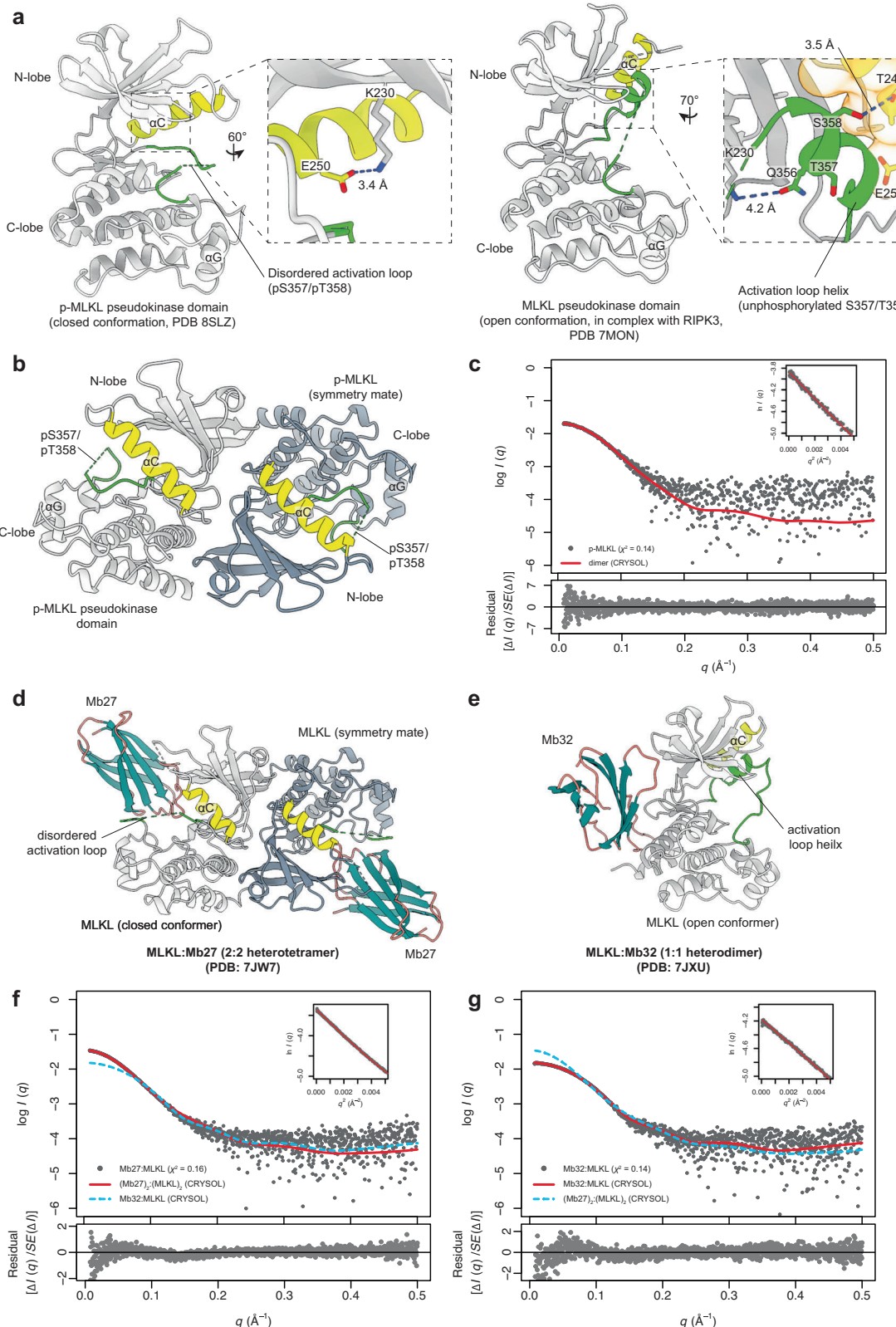

**a**

p-MLKL pseudokinase domain
(closed conformation, PDB 8SLZ)

MLKL pseudokinase domain
(open conformation, in complex with RIPK3,
PDB 7MON)

**b**

p-MLKL pseudokinase domain

**c**

**d**

MLKL:Mb27 (2:2 heterotetramer)
(PDB: 7JW7)

**e**

MLKL:Mb32 (1:1 heterodimer)
(PDB: 7JXU)

**f**

**g**

phosphorylation would disrupt a hydrogen bond with T246 of the αC helix to likely favour activation loop mobility and solvent exposure.

## Closed conformation drives human MLKL pseudokinase domain dimerization

Analysis of all 13 human MLKL pseudokinase domain structures reported to date, in which MLKL adopts the closed conformation, revealed that human MLKL pseudokinase domain dimerized in crystallo via the same hinge interface in all crystal structures, regardless of crystal form (Table 2 and Supplementary Table 3). In contrast, when human MLKL pseudokinase domain was crystallized in the open conformation (PDB: 7JXU and 7MON)[29,33], a dimer was not observed in these crystal lattices (Supplementary Fig. 2c). This trend strongly supports the idea that switching into the closed

**Fig. 1 | RIPK3-mediated phosphorylation drives human MLKL pseudokinase domain dimerization by promoting the closed conformation. a** Human MLKL pseudokinase domain (grey ribbon; yellow, αC helix; green, activation loop) undergoes a conformational change upon phosphorylation of the activation loop (T357/S358). Left, crystal structure of p-MLKL pseudokinase domain adopts a closed conformation with an intact VAIK-αC salt bridge (zoomed panel) and disordered activation loop. Right, previously published structure of MLKL pseudokinase domain in complex with RIPK3 kinase domain[29] adopts an open conformation with a disrupted VAIK-αC salt bridge (zoomed panel) and a structured, helical activation loop where T357/S358 are not phosphorylated. Zoomed panel, unphosphorylated S358 is buried underneath the αC helix (orange transparent surface) and forms a hydrogen bond with T246 of the αC helix. **b** Human phosphorylated (p)-MLKL pseudokinase domain dimer model generated from the crystallographic symmetry mate (symmetry mate coloured slate grey). **c** Small angle X-ray scattering (SAXS) scattering profile (grey points) of human p-MLKL pseudokinase domain support the pseudokinase domain dimer model. Theoretical scattering profile of

p-MLKL pseudokinase domain dimer was calculated using CRYSOL and shown as red line. Guinier plot is shown as the insert. Standardized residual plot comparing theoretical and experimental scattering profiles is shown on the bottom. **d** Mb27 (teal and salmon) binding stabilizes the closed MLKL pseudokinase domain (grey ribbon; yellow, αC helix; green, activation loop) conformation and promotes pseudokinase domain dimerization in solution. Model of MLKL:Mb27 pseudokinase domain complex in a 2:2 stoichiometry generated from the symmetry mate of MLKL:Mb27 co-crystal structure (PDB: 7JW7; symmetry mate of MLKL coloured slate grey)[33]. **e** Mb32 (teal and salmon) binding stabilizes the open MLKL pseudokinase domain conformation and stabilizes monomeric pseudokinase domain in solution. Crystal structure of Mb32:MLKL pseudokinase domain heterodimeric complex (PDB: 7JXU)[33]. **f, g** SAXS scattering profiles (grey points) of Monobody:MLKL complexes (**c** MLKL:Mb27; **d** MLKL:Mb32) show strong concordance with the theoretical scatter (red) calculated from their respective complex models. The theoretical scattering curves from (**f, g**) are included for comparison in (**g, f**), respectively, as blue dashed lines.

conformation occurs concurrently with pseudokinase domain dimerization.

We sought to test whether stabilizing the closed conformation in vitro using Monobodies would promote MLKL pseudokinase domain dimerization. In our previous study, co-crystal structures revealed that Mb27 binds the closed conformer of human MLKL pseudokinase domain (PDB: 7JW7) (Fig. 1d), whereas Mb32 binds MLKL in an open conformation (PDB: 7JXU) (Fig. 1e)[33]. The epitope of Mb27 is

### Table 1 | SAXS data collection and analysis statistics

| Data collection parameters | | | |
|---|---|---|---|
| Instrument | Australian Synchrotron SAXS/WAXS beamline | | |
| Detector | PILATUS3-2M (Dectris) | | |
| Wavelength (Å) | 1.0332 | | |
| $q$ range (Å⁻¹)ᵃ | 0.005–0.5 | | |
| Maximum flux at sample | $8 \times 10^{12}$ photons/s at 12 keV | | |
| Temperature (K) | 283 | | |
| Exposure time | Serial 1 s exposures | | |
| Sample configuration | SECS-SAXS with co-flow | | |
| Flow rate | 0.2 mL/min, Superdex-200 Increase 5/150 column | | |
| Inline gel filtration chromatography buffer | 0.2 M NaCl, 20 mM HEPES pH 7.5, 5% v/v glycerol | | |
| | **p-MLKL pseudo-kinase (190–471)** | **MLKL pseudokinase: Mb27** | **MLKL pseudokinase: Mb32** |
| Protein concentration | 40 μL of 7.2 mg/mL | 50 μL of 5.6 mg/mL | 50 μL of 5.6 mg/mL |
| **Structural parameters** | | | |
| $I(0)$ (cm⁻¹) (from Guinier) | $0.021 \pm 0.00028$ | $0.035 \pm 0.00013$ | $0.015 \pm 0.000078$ |
| $R_g$ (Å) (from Guinier) | $27.59 \pm 0.62$ | $31.09 \pm 0.20$ | $22.42 \pm 0.20$ |
| $I(0)$ (cm⁻¹) [from $P(r)$] | $0.021 \pm 0.00030$ | $0.035 \pm 0.00011$ | $0.015 \pm 0.000057$ |
| $R_g$ (Å) [from $P(r)$] | $27.58 \pm 0.02$ | $31.58 \pm 0.03$ | $22.44 \pm 0.02$ |
| $D_{max}$ (Å) | 89.71 | 105.37 | 64.45 |
| **Software employed** | | | |
| Primary data reduction | Scatterbrain (Australian Synchrotron), CHROMIXS | | |
| Data processing | PRIMUSQT, GNOM | | |
| Computation of model intensities | CRYSOL | | |

ᵃ$q$ is the magnitude of the scattering vector, which is related to the scattering angle ($2\theta$) and the wavelength ($\lambda$) as follows: $q = (4\pi/\lambda)\sin\theta$.

located away from the dimerization interface of MLKL, strongly suggesting that Mb27 does not sterically interfere with MLKL dimerization. In contrast, Mb32 binds the human MLKL pseudokinase domain hinge at a site that overlaps the back-to-back dimerization interface; thus, MLKL dimerization would occlude Mb32 binding. Accordingly, these Monobodies can be used as tools to stabilize the distinct conformations of the human MLKL pseudokinase domain in solution. We generated recombinant protein complexes between the human MLKL pseudokinase domain and these Monobodies and examined their structures in solution using SEC-SAXS. The sizes of the MLKL pseudokinase domain in complex with different Monobodies were evidently different in solution (Table 1 and Fig. 1f, g). The $R_g$, $D_{max}$ and scattering profile of the Mb32:MLKL pseudokinase domain complex are consistent with the 1:1 heterodimeric complex observed in the crystal structure, while the Mb27:MLKL pseudokinase domain complex was consistent with a 2:2 heterotetrameric complex (Table 1 and Fig. 1f, g). In the crystal structure of the Mb27:MLKL complex, the MLKL pseudokinase domain forms a dimer with its symmetry mate via the same interface as the p-MLKL pseudokinase domain (Fig. 1d), supporting the idea that Mb27 binding promotes the formation of a 2:2 heterotetrameric complex with MLKL pseudokinase domain in solution. This shows that when the pseudokinase domain of human MLKL is stabilized in the closed conformation by Mb27, it dimerizes in solution in the same manner as p-MLKL pseudokinase domain. These data indicate that the closed conformation is coupled to human MLKL pseudokinase domain dimerization.

### Pseudokinase domain dimerization drives full-length human MLKL tetramerization

We next examined whether pseudokinase domain dimerization plays a functional role in the context of full-length human MLKL by performing biophysical analyses on recombinant full-length human MLKL co-expressed with either RIPK3 or Monobodies in *Sf*21 insect cells. When wild-type recombinant full-length human MLKL is expressed alone in insect cells, MLKL elutes primarily as monomers, with a smaller proportion eluting as tetramers[40]. These tetramers are considered to represent the activated, pro-necroptotic form of human MLKL. Remarkably, when we co-expressed wild-type full-length MLKL with the kinase domain of RIPK3 (residues 1–316), full-length MLKL eluted predominantly as tetramers on size-exclusion chromatography (Fig. 2a), presumably driven by phosphorylation mediated by co-expressed RIPK3 in cells. Similarly, by promoting pseudokinase domain dimerization, co-expression with Mb27 allowed all full-length MLKL to form tetramers in solution (Fig. 2a). Mass spectrometry revealed that the MLKL tetramer from co-expression with RIPK3 is disulfide-linked via C184, including when 0.5 mM TCEP was used in lysis and IMAC purification steps (Supplementary Fig. 3a). The biological function of this crosslink is yet to be studied, although C184

**Table 2 | Summary of all crystal structures of MLKL pseudokinase domain to date**

| PDB | Species | Variant | Ligand | Residue range | Space group[a] | Conformation | | | Reference |
|-----|---------|---------|--------|---------------|----------------|--------------|---|---|-----------|
| | | | | | | Salt bridge (VAIK-αC) | R-spine | DFG | |
| 4MWI | Human | – | – | 190–471 | C 2 2 2₁ | Yes | Yes | In | 42 |
| 8SLZ | Human | pT357/pTS358 | – | 190–471 | C 2 2 2₁ | Yes | Yes | In | This study |
| 4M67 | Human | – | – | 179–471 | C 2 2 2₁ | Yes | Yes | In | 54 |
| 6LK6 | Human | T357A/S358A | – | 179–471 | C 2 2 2₁ | Yes | Yes | In | 52 |
| 6LK5 | Human | T357E/S358D | – | 179–471 | C 2 2 2₁ | Yes | Yes | In | 52 |
| 6BWK | Human | T357E/S358E | – | 190–471 | C 2 2 2₁ | Yes | Yes | In | 40 |
| 5KO1 | Human | – | Compound 4 (Type I inhibitor) | 191–471 | C 2 2 2₁ | Yes | Yes | In | 55 |
| 5KNJ | Human | – | Compound 1 (Type II inhibitor) | 191–471 | P 2₁ 2₁ 2₁ | Yes | No | Out | 55 |
| 6O5Z | Human | – | Compound 2 (Type II inhibitor) | 190–471 | P 2₁ | Yes | No | Out | 56 |
| 7JW7 | Human | – | Mb27 (protein) | 190–471 | P 3₁ 2 1 | Yes | Yes | In | 33 |
| 7JXU | Human | – | Mb32 (protein) | 190–471 | P 4₃ | No | No | In | 33 |
| 7MON | Human | – | RIPK3 (protein) | 190–471 | P 2₁ 2₁ 2₁ | No | No | In | 29 |
| 4BTF | Mouse | – | – | 1–464 | P 2₁ 2₁ 2₁ | No | No | In | 32 |
| 4M69 | Mouse | – | RIPK3 (protein) | 182–464 | C 2 2 2₁ | No | No | Out | 54 |
| 4M68 | Mouse | – | – | 182–464 | P 2₁ | No | No | In | 54 |
| 6VC0 | Horse | – | – | 188–475 | C 2 | Yes | Yes | In | 36 |
| 6VBZ | Rat | – | – | 179–464 | P 6₁ 2 2 | Yes | Yes | In | 36 |

[a]Details of unit cell dimensions are provided in Supplementary Table 3.

crosslinks may provide a basis for the tetramers that have been observed by non-reducing PAGE and correlated with MLKL activation[47]. Human MLKL C184 is highly conserved among primate and bird MLKL orthologs. However, this cysteine is not conserved in rodent MLKL orthologs, where other cysteines have been implicated in mouse MLKL oligomerization[58]. A faint band on SDS-PAGE corresponding to the RIPK3 kinase domain co-eluted with full-length p-MLKL tetramer (Fig. 2b), suggesting that this p-MLKL tetramer is bound sub-stoichiometrically by RIPK3. The sub-stoichiometric binding of RIPK3 is consistent with the previous finding that RIPK3 disengages from phosphorylated MLKL[33]. In contrast, when the phosphorylation sites of MLKL were mutated to T357E/S358E to prevent their phosphorylation, an intense RIPK3 band co-elutes with monomeric full-length MLKL (Fig. 2a, b), consistent with the formation of a 1:1 stoichiometric complex. Furthermore, these data suggest that the T357E/S358E phosphomimetic mutations incompletely capture the properties of phosphorylation required to promote MLKL oligomerization and dissociation from RIPK3.

The Monobody, Mb32, binds to the hinge of the pseudokinase domain and thus overlaps with the dimerization interface (Fig. 2c)[33]. As such, we used Mb32 as a tool to examine whether the pseudokinase domain dimerizes in the context of the full-length human MLKL tetramer. When a recombinant Mb32 bearing an N-terminal His₆-tag (His-Mb32) was mixed with the wild-type human p-MLKL tetramer purified following co-expression with RIPK3 kinase domain, negligible amounts of MLKL could be pulled down with His-Mb32 by Ni-NTA (Fig. 2d, e). In contrast, His-Mb27, whose epitope does not overlap with the pseudokinase domain dimerization interface, could pulldown wild-type p-MLKL tetramer. As expected, the inactive T357E/S358E MLKL:RIPK3 heterodimer co-precipitated with His-Mb32 but not His-Mb27, consistent with our earlier prediction that the Mb27 binding site on MLKL would be occluded by RIPK3 in this complex[33]. As expected, a control Monobody, His-Mb33, which binds the N-terminal 4HB domain[59], could co-precipitate MLKL regardless of oligomeric state. These data support pseudokinase domain dimerization occurring within the full-length human MLKL tetramer in solution via the interface observed in

the dimeric p-MLKL pseudokinase domain crystal structure. Our findings with recombinant proteins were mirrored in HT29 cells undergoing necroptosis, where Mb27-bound MLKL was assembled into oligomers on BN-PAGE (Fig. 2f), while Mb32-bound MLKL was primarily monomeric. Collectively, these data suggest that RIPK3-mediated phosphorylation drives MLKL oligomerization by switching the pseudokinase domain conformation, which prompts pseudokinase domain dimerization and protrusion of the brace helix to enable tetramerization.

**Full-length MLKL tetramers assemble via interfaces on all three component domains**

To gain insights into how the pseudokinase domain controls the oligomeric state of full-length human MLKL, we sought to visualize the structure of the recombinant full-length human MLKL tetramer. Although efforts to determine this structure by cryo-electron microscopy have thus far been hindered by extensive protein aggregation at the air–water interface, and significant flexibility in the N-terminal region of the tetramer, we could acquire structural insights using a combination of computational modelling and negative stain electron microscopy.

We first sought to generate a structure prediction of full-length human MLKL tetramer using ColabFold-Multimer[60]. Due to computational limitations, we generated two tetrameric predictions using truncated human MLKL sequences. The first model was generated using residues 1–179, which includes the N-terminal 4HB domain (residue 1–120) and the entire brace region (residue 121–179) (Fig. 3a, left). The second model was generated from residue range 147–471, which includes the C-terminal region of the brace (residue 147–179) and the pseudokinase domain (Fig. 3a, left). The C-terminus of the brace region is present in both models, and the structural predictions are highly consistent across the two models (RMSD = 0.279 Å over 110 Cα atoms), which allowed us to use this region to align and merge two models into a single model of full-length human MLKL tetramer (Fig. 3a, middle). Interestingly, the pseudokinase domain dimers within this model were predicted by ColabFold to form the back-to-back

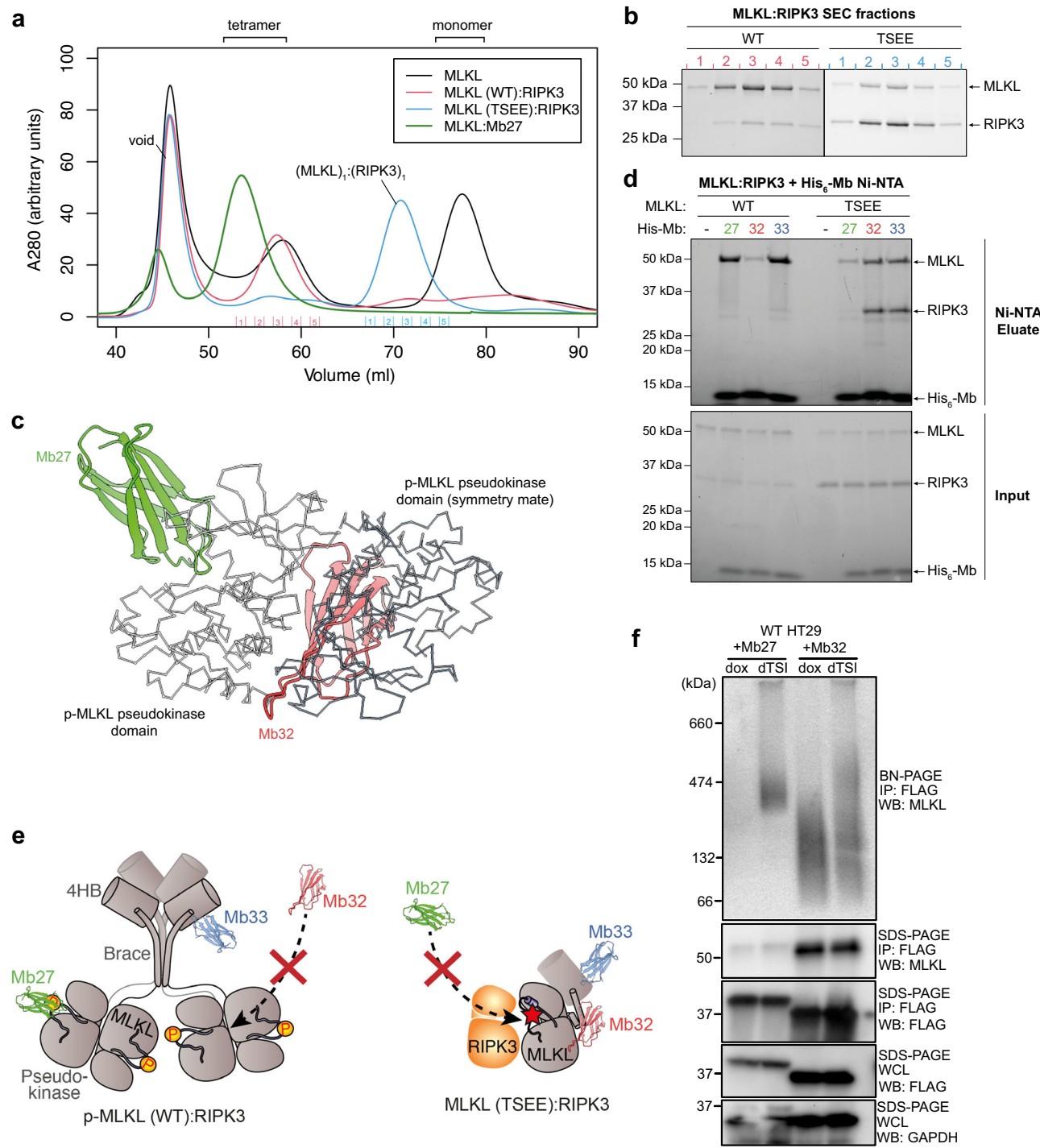

dimer observed in our p-MLKL pseudokinase domain structure, albeit in the open kinase conformation with a helical activation loop and displaced αC helix. This likely represents a conformational bias of ColabFold towards a conformation with fewer unstructured regions. Because the closed conformation was experimentally observed in our p-MLKL pseudokinase domain structure (Fig. 1b), we replaced the ColabFold modelled domains with the experimentally-determined crystal structure of the dimeric pseudokinase domain. We linked the brace helices and the dimeric pseudokinase domains using the Rosetta Fast Relax protocol[61] to optimize the composite model and eliminate sidechain clashes introduced by ColabFold (Fig. 3a, right). The relaxed model of full-length MLKL tetramer predicts that the four protomers assemble into an elongated tetramer, where they interact via three interfaces between: 4HB domains, brace helices, and within each pair

of pseudokinase domains (Fig. 3a, b). The brace helices are connected to the other domains via disordered linker regions with low pLDDT scores, perhaps indicating that they are likely flexible in solution. Overall, the interactions within the pseudokinase domain dimer and the C-terminal region of the brace helices are well supported by high pLDDT scores and existing knowledge of MLKL biology, although the precise interactions within the 4HB domain and the N-terminal region of the brace helices are less well defined.

We next sought to experimentally validate the ColabFold full-length human MLKL tetramer model using negative stain electron microscopy. The tetramer peak from size-exclusion chromatography of the recombinant full-length human MLKL was used for this experiment (Fig. 2a, black line). We observed 2D class averages with size and shape that are consistent with the elongated tetrameric model

**Fig. 2 | Pseudokinase domain dimerization drives full-length MLKL tetramerization in solution. a** Size-exclusion chromatograms of full-length human MLKL [red, wild-type (WT); green, T357E/S358E (TSEE) mutant] in complex with RIPK3 kinase domain, overlaid with the chromatogram of MLKL alone (black), and MLKL tetramer bound by Mb27 (blue). Fractions analysed by SDS-PAGE are labelled on the x-axis [red, MLKL (WT):RIPK3; blue, MLKL (TSEE):RIPK3]. **b** Stain-free reducing SDS-PAGE of full-length human MLKL:RIPK3 kinase complexes size-exclusion chromatography fractions. Data in (**a**, **b**) are representative of two independent repeats. **c** Mb32 (red) cannot bind the MLKL pseudokinase domain due to binding epitope overlap with the dimerization interface, while Mb27 (green) binds the dimer by binding to the ATP-binding cleft. Crystal structures of Mb32:MLKL pseudokinase domain (PDB: 7JXU; MLKL not shown) and Mb27:MLKL pseudokinase domain (PDB: 7JW7; MLKL not shown)[33] were aligned with the crystal structure of p-MLKL pseudokinase domain dimer (PDB: 8SLZ; αC trace, white and slate grey) using UCSF ChimeraX. **d, e** Pseudokinase domain dimer is detected within p-MLKL tetramer (from co-expression with RIPK3 kinase domain), but not within the TSEE

mutant MLKL:RIPK3 heterodimer. **d** Binding of N-terminal His₆-tagged Monobodies to recombinant MLKL:RIPK3 complexes were examined by Ni-NTA precipitation and stain-free SDS-PAGE. Mb32 is unable to bind wild-type p-MLKL tetramer, confirming that pseudokinase domain dimerizes within this tetramer. Mb27 is able to bind p-MLKL tetramer, but not TSEE mutant MLKL:RIPK3 heterodimer, because RIPK3 hinders the Mb27 epitope[29,33]. Control Mb33[59] binding to the 4HB domain is not affected by MLKL oligomeric state. Data represent two independent replicates. **e** Illustration of the mechanisms by which Monobodies detect pseudokinase domain dimerization in full-length MLKL complexes. Richardson (Ribbon) diagrams of Mb27 (from PDB: 7JW7)[33], Mb32 (from PDB: 7JXU)[33] and Mb33 (from PDB: 6UX8)[59] were drawn using PyMol. **f** Mb27 binds exclusively MLKL tetramer upon necroptotic stimulation in HT29 cells. HT29 cells stably expressing Monobodies[33] were harvested at 4.5 h post necroptosis stimulation by TSI and subjected to FLAG-immunoprecipitation. The FLAG-eluates were analysed on Blue Native (BN)-PAGE or SDS-PAGE and immunoblotting with antibodies indicated. Data representative of two independent replicates.

generated using ColabFold (Fig. 3c). These 2D class averages typically display a bi-lobal shape connected to a narrow stalk in the middle, which likely correspond to two pseudokinase domain dimers connected to the brace helices. The density corresponding to the N-terminal 4HB bundle domains is poorly-defined, consistent with 4HB domain flexibility in solution and the low model pLDDT score of this region in the ColabFold model. Some 2D class averages also display a tri-lobal shape, with two larger densities and one smaller density. These could represent a bottom-up view of these particles, where the smaller density can be attributed to the brace helices and possibly N-terminal 4HB domains. Similar features were observed in the 2D classes of the full-length MLKL tetramer stabilized by Mb27 (Supplementary Fig. 4a).

Overall, this model is consistent with existing functional studies on MLKL tetramerization[37,40,43,47,52,59,62]. Unexpectedly, however, while the pseudokinase domains dimerize within the model of human MLKL tetramer, the two dimers are not predicted to be in contact with each other and are connected to the brace helices via disordered linkers (Fig. 3b). The pseudokinase domain alone is insufficient for tetramerization, and this is consistent with previous reports that the brace helices are essential for MLKL oligomerization[37,47]. Unlike the crystal structure of the mouse MLKL monomer (PDB: 4BTF)[32] where the brace region forms two α-helices at a sharp angle, the entire brace region forms one single elongated helix spanning ~80 Å. The brace helices from four protomers assemble into a coiled-coil at the centre of this tetramer. Consistent with previous sequence-based prediction using Multicoil[47], the coiled-coil of brace helices in this model is mediated largely by a core of hydrophobic sidechains (F148, L151, I158, L162, L165, M169, I172, L176) positioned in a 4-fold rotationally symmetric arrangement (Fig. 3d). Notably, C184 residues are in close proximity to each other on the flexible loops between the brace helix coiled coil and the pseudokinase domain (Supplementary Fig. 3b), which explains their propensity to form disulfide bonds in recombinant human MLKL tetramer (Supplementary Fig. 3a).

The position of the N-terminal portion of the brace helix (residue 123–154) in this tetrameric model is distinct from all previously reported crystal (PDB: 6UX8, 6ZVO)[59,63] or solution NMR (PDB: 6D74, 2MSV, 6ZPR, 6ZLE, 7NM4, 7NM5, 7NM2)[43,63–65] structures of the 4HB domain with a portion of the brace region that corresponds to the first of two brace helices in the full-length mouse MLKL structure[32]. In the earlier human MLKL 4HB-brace structures, the first brace helix packs against the helices α2 and α4 from the 4HB domain (Fig. 3e, f). In contrast, in the tetramer model, the 4HB domain is released from the brace helix (Fig. 3g, h), allowing the four 4HB domains to interact with each other, forming a 4-fold rotationally symmetric bundle of 4HB domains (Fig. 3b, g). At the core of this bundle, the side chains of Q117 from four protomers are positioned to form hydrogen bonds that would stabilize the tetrameric arrangement (Supplementary Fig. 4b).

The α4 helix of one 4HB domain inserts into the grove between the α2′ and α4′ helices of the adjacent protomer (Fig. 3g, h), mediated by salt bridges between α4 (D100, D107) and α2′ helix residues (K26, R30, R34) and hydrophobic interactions between α4 (L114, V118) and a groove at the C-terminal ends of the α2′ (L38, P41, L45) and α4′ (W109, L116) helices (Fig. 3h). Importantly, the interaction interface formed by the α2 and α4 helices is the same interface that is occupied intramolecularly by the brace helix in previously reported autoinhibited structures (Fig. 3e, f). This suggests that the brace regions' role in the steady state is to occupy the 4HB α2-α4 helix groove, and thereby inhibit the formation of the permeabilization-competent 4HB tetramer observed in our full-length model. These observations rationalize the previously proposed 'plug-release' mechanism of the brace helix[43,49], and compellingly, the 4HB tetramer in our model forms a central, large, positively-charged pocket which we hypothesize would interact with negatively-charged headgroups of phosphoinositol phosphates reported to bind MLKL[40,48–50] (Fig. 3g). Intriguingly, a more recent version of ColabFold Multimer predicts an alternative 4HB domain conformation using the same sequence (residues 1–179) (Supplementary Fig. 4c). Although the C-terminal region of the brace helix (residues 153–179) consistently forms a coiled coil, the N-terminal portion (residues 130–152) remains bound to the 4HB domain like that in previously reported structures of the autoinhibited, "plugged" conformation[43]. This alternative model raises the possibility that a permeabilization-incompetent MLKL tetramer may assemble prior to the 4HB domain's release from the brace region. Taken together, these models predict that full-length MLKL tetramerization involves both pseudokinase domain dimerization and the assembly of the extended brace helices into a coiled coil, which allows the unleashing and assembly of 4HB domains into a membrane disrupting conformation.

## Pseudokinase domain-driven MLKL tetramerization unleashes the 4HB domain executioner function

We next sought to examine the role of pseudokinase domain-driven human MLKL tetramerization in membrane permeabilization and cell death. First, we tested the abilities of recombinant full-length MLKL complexes to permeabilize liposomes in vitro by measuring the release of a fluorescent, self-quenching dye. Tetrameric wild-type MLKL rapidly permeabilized liposomes, as detected by fluorescent dye release, whereas negligible liposome permeabilization was detected for monomeric MLKL (Fig. 4A). The permeabilization activity of MLKL tetramers was enhanced modestly by phosphorylation of the pseudokinase domain activation loop when co-expressed in *Sf*21 cells with RIPK3, but even more profoundly by stabilization with Mb27 (Fig. 4A). In contrast, the monomeric T357E/S358E MLKL complexed with RIPK3 exhibited almost no liposome permeabilization activity. These data support the idea that tetrameric human MLKL, which assembles upon pseudokinase domain

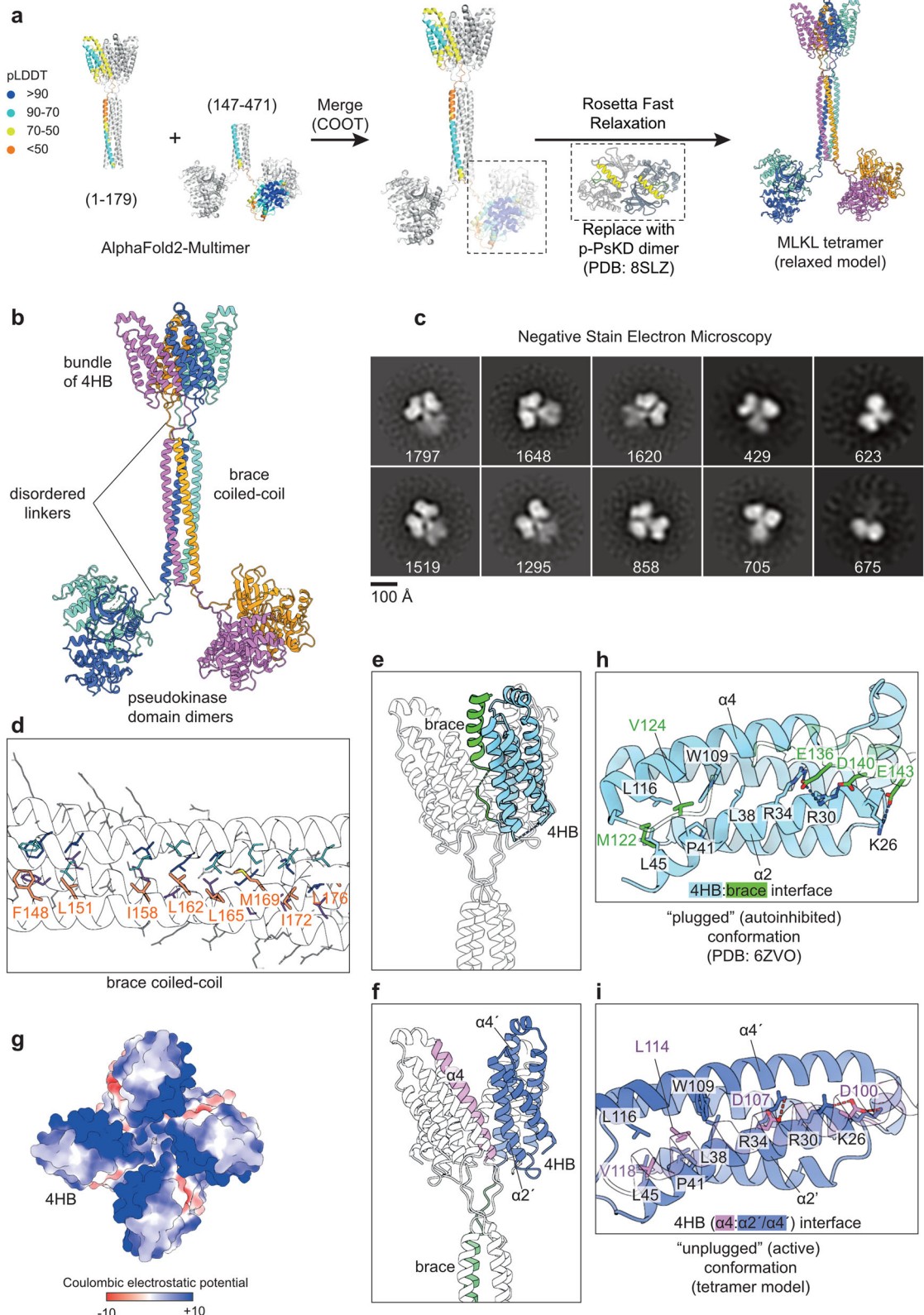

dimerization, is the form of MLKL capable of mediating membrane disruption during necroptosis.

Our tetrameric model suggests that release of the 4HB from the brace region allows formation of a permeabilization-competent 4HB domain tetramer (Fig. 3g, h), consistent with previous reports that the brace helices attenuate the membrane permeabilization function of

4HB domain prior to MLKL activation[37,47]. To validate whether release of the brace helix from the 4HB domain during MLKL tetramerization acts as a regulatory mechanism to unleash the cell killing activity of 4HB domain, we generated a recombinant human MLKL 4HB domain (residues 2–123) without the brace helix (Supplementary Fig. 5a) by introducing a HRV3C protease cleavage site in the flexible loop

**Fig. 3 | Full-length human MLKL assembles into an elongated tetramer via three interfaces. a** Generation of the model of full-length human MLKL tetramer. Two truncated sequences (residues 1–179 and 147–471) were used in ColabFold-Multimer to generate two tetrameric predictions which overlap via the brace helices coiled coil. The overlapping region was used to align the two models and merge them together, and the pseudokinase domains were replaced with experimentally determined structures of phosphorylated (p)-MLKL pseudokinase domain dimer, before free energy minimization using Rosetta Fast Relax protocol to remove sidechain steric clashes. **b** Final relaxed model of human MLKL tetramer coloured by chain. **c** 2D classes from negative stain electron microscopy on full-length human MLKL tetramer. The number of particles in each class is shown in white text. **d** An array of hydrophobic residues (thick sticks, coloured according to panel b) mediate the formation of brace helix coiled coil. Peptide backbones are shown as silhouette. **e, f** The brace helix is 'unplugged' from the four-helix bundle

(4HB) domain in the full-length MLKL tetramer. The model of full-length MLKL tetramer is shown as a silhouette. **e** Previously reported crystal structure of the 4HB domain (cyan) with the brace helix (green) in an autoinhibited conformation is overlaid with the MLKL tetramer model. **f** The brace helix (green) is pulled away from the 4HB domain (blue) in the full-length MLKL tetramer. Helix α4 from the adjacent protomer (purple) occupies the groove between the α2' and α4' helices, the same site where the brace helix binds in the inactive conformation. **g** The formation of 4HB domain tetramer exposes a large positively charged pocket. Top down view of the 4HB domains within the MLKL tetramer model shown as a surface coloured according to Coulombic electrostatic potential. **h, i** The α2 and α4 helices of the 4HB domain (opaque ribbon; **h** cyan; **i** blue) mediates both the interaction with the brace helix (**h** green transparent ribbon) and the interaction with helix α4 (**i** purple transparent ribbon) from the adjacent subunit in a MLKL tetramer. Residues involved in interactions are shown as sticks and labelled.

between the 4HB domain (residues 2–123) and brace helix (residues 130–154). Upon cleaving off the brace helix, the human 4HB domain more rapidly permeabilized liposomes than the full-length MLKL tetramer stabilized by Mb27 (Fig. 4B). In contrast, human 4HB domain without cleaving off the brace helix (2–154) exhibited negligible liposome permeabilization activity, consistent with previous reports[41,43,49] (Fig. 4B).

We next sought to examine the contributions of the MLKL tetramerization interface residues to the necroptotic functions of human MLKL in HT29 cells through mutational analyses. We substituted interface residues with Asp or Glu mutations to disrupt interfaces by introducing negative charge in place of basic Arg or Lys residues, or charge in place of hydrophobic or uncharged sidechains. We stably reconstituted *MLKL*^-/- HT29 cells with doxycycline-inducible wild-type or interface mutant MLKL constructs via lentiviral vectors (Supplementary Fig. 5b), and tested the ability of each construct to signal for necroptotic cell death. Exogene expression was induced with doxycycline, and the capacity of cells to undergo necroptosis was assessed in the absence or presence of the necroptotic stimulus (TSI: T, TNF; S, Smac mimetic Compound A; I, pan-Caspase inhibitor IDN-6556/Emricasan). Cell death was monitored by SYTOX Green uptake and total cells were enumerated by DRAQ5, using IncuCyte live cell imaging[66] (Fig. 4C and Supplementary Fig. 5c). Mutations disrupting the interfaces between 4HB domains (R34E, Q117E; Figs. 3h and 5a) and the brace region coiled-coil (L176D, I158E, M169E, L172D; Figs. 3d and 5b) completely abolished necroptotic cell death (Fig. 4C). Mutation of MLKL Q120, which sits on the periphery of the inter-4HB domain interfaces (Fig. 5b), and F266, which projects from the pseudokinase domain hinge to contribute to the dimerization interface (Fig. 5c), attenuated necroptotic cell death. These data implicate these interfaces as partial contributors to the assembly of pro-necroptotic MLKL oligomers in human cells. In contrast, mutation of human MLKL C184 did not compromise necroptosis, despite our finding that C184 forms intermolecular disulfide bonds in recombinant human MLKL tetramers (Supplementary Fig. 3). This finding is reminiscent of observations reported for mouse MLKL, where mutation of Cys residues resulted in defective oligomerization without compromising necroptotic cell death signalling[58]. Consistent with defective necroptotic signalling, Q117E, I158E, L176D, L172D and M169D human MLKL mutants failed to assemble into oligomers upon necroptotic stimulation when examined in BN-PAGE immunoblots (Fig. 4D). Intriguingly, R34E human MLKL assembled into higher molecular weight oligomers and translocated to biological membranes, despite not possessing membrane permeabilization capacity (Fig. 4C, D). The human MLKL tetramer model predicts that R34 forms a salt-bridge with D107. This is supported by the observation that, like R34E, D107A/E111A human MLKL also oligomerises and translocates to membranes, but does not kill when expressed in human *MLKL*^-/- U937 cells[40]. Our mutational data and those reported by others[43,47,49] strongly support that the MLKL tetramer conformation presented in Fig. 3 corresponds to the active,

permeabilisation-competent conformation of MLKL, and suggests that if the alternative conformation generated by ColabFold in which the brace latches the 4HB domain can form, it must be a precursor to the active form (Supplementary Fig. 4c).

Consistent with our prediction that the R30E mutation would disrupt salt bridges between the 4HB domain (R30) and the brace region (E136 and D140) (Fig. 3f), expression of R30E human MLKL resulted in constitutive cell death upon expression in *MLKL*^-/- HT29 cells, in the absence of any stimulus (Fig. 4C) and independently of RIPK3-mediated phosphorylation (Supplementary Fig. 5b). Following expression alone, the human MLKL R30E mutant oligomerises into higher order structures that associate with biological membranes (Fig. 4D). This result is consistent with earlier reports of recombinant K26E/R30E 4HB domain plus brace region (residues 2–154) enhancing liposome permeabilization[43], and introduction of R30E full-length human MLKL along with oligomerizable human RIPK3 into mouse *Mlkl*^-/-/*Ripk3*^-/- fibroblasts exhibiting hyperactive killing function[49]. Coupled with our findings, these data strongly support the release of the 4HB domain from the brace helix when MLKL assembles into a membrane-disrupting tetramer as a critical step in MLKL activation. Constitutive cell death mediated by MLKL R30E in the absence of stimulus is not inhibited by the 4HB domain-targeting MLKL covalent inhibitor, necrosulfonamide (NSA)[30], but partially inhibited by RIPK3 kinase inhibitor, GSK´872[23] (Fig. 4E). Treatment of *MLKL*^-/- HT29 cells expressing MLKL R30E with the TSI necroptotic stimulus led to cell death of magnitude and kinetics comparable to wild-type MLKL-expressing control cells. However, compared to cells expressing wild-type MLKL, R30E MLKL was less susceptible to the MLKL inhibitor, NSA, and more potently inhibited by the RIPK3 inhibitor, GSK'872, in the presence of TSI (Fig. 4E). Our functional studies of recombinant MLKL and MLKL mutants in human cells validate the importance of interactions between the three component domains in MLKL in the assembly of the pro-necroptotic MLKL tetramer, and provide important validation for the elongated MLKL tetramer model.

## Discussion

Phosphorylation of the MLKL pseudokinase domain by RIPK3 kinase has long been established as a crucial checkpoint in the necroptosis pathway[30,32]. However, the precise mechanism by which RIPK3-mediated phosphorylation drives necroptotic cell death is incompletely understood. Here, we identify RIPK3-mediated phosphorylation as the cue for the MLKL pseudokinase domain to switch into a closed conformation which favours pseudokinase domain dimerization. This unleashes the brace region into an elongated helix, which tetramerizes by assembling into a coiled coil. The formation of this coiled coil may facilitate 4HB domain release from the brace helices, enabling the organization of the four 4HB domains into an assembly capable of membrane permeabilization (Fig. 5a–d).

MLKL oligomerization is essential for necroptosis, but the precise structural nature of the full-length MLKL oligomer has been a matter of

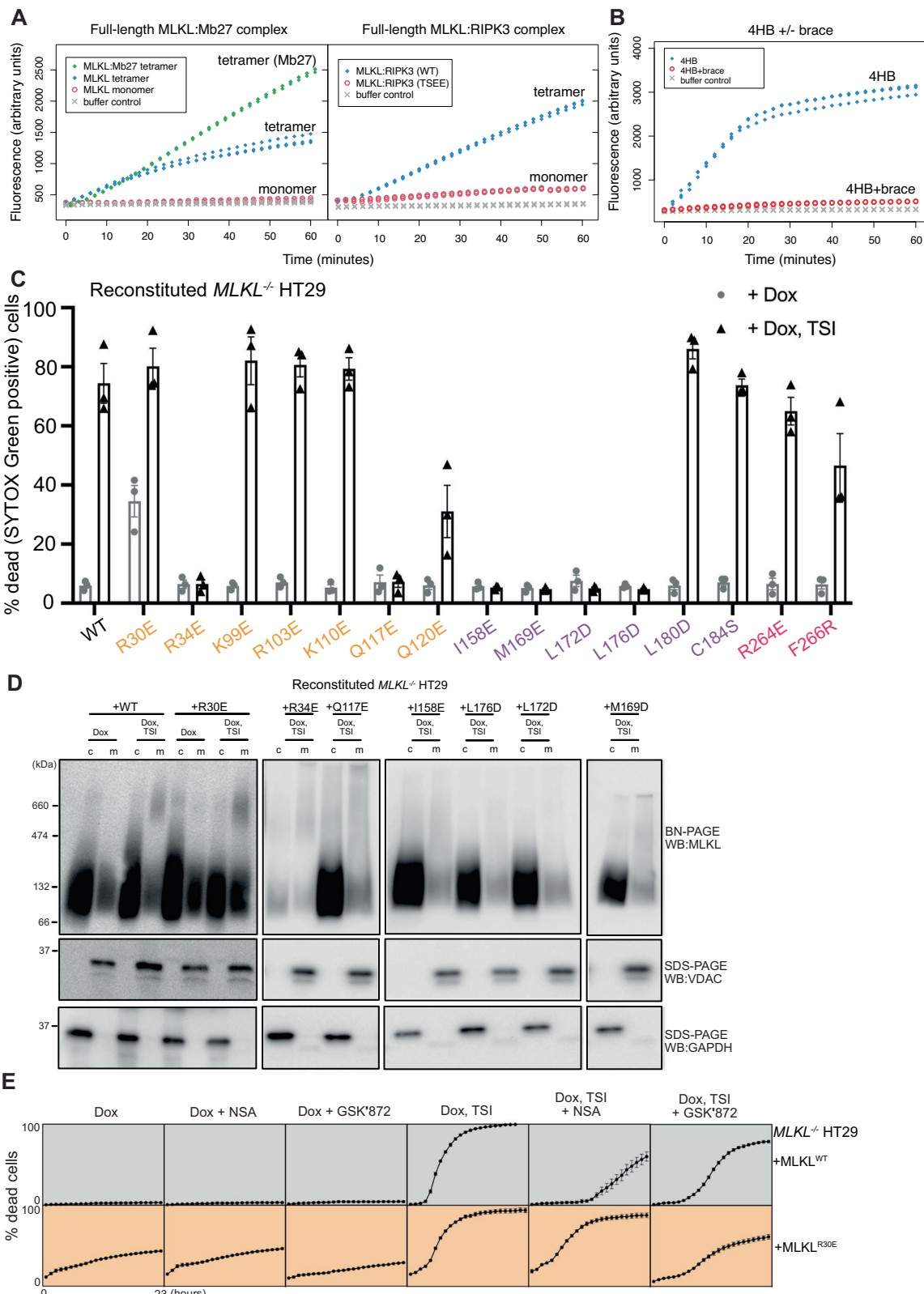

debate[37–40,58,67,68], because high resolution structural studies of human MLKL have been limited to studies of the component domains. Here, we used negative stain electron microscopy to validate a structural prediction of full-length human MLKL tetramer generated using ColabFold. This model yielded crucial mechanistic insights into human MLKL and rationalizes previously reported biochemical data. Consistent with its proposed role for driving MLKL tetramerization,

disruption of pseudokinase domain dimerization interface impaired (F266R, this study) or abolished (H223A/R224A/E258A/P260G/R264A/F266A)[52] necroptosis (Fig. 5c). Previous studies have established the critical role of the brace region in mediating MLKL oligomerization, because mutations in this region also abolish oligomerization and necroptotic cell death[37,47]. The formation of a coiled coil by the brace region had been predicted based on the positions of hydrophobic

**Fig. 4 | MLKL tetramerization is essential for necroptotic cell death.**
**A, B** Liposome dye release assays on recombinant human MLKL protein complexes. **A** Tetrameric MLKL (unliganded, stabilized by Mb27, or phosphorylated by RIPK3) are considerably more effective at permeabilizing liposomes than monomeric MLKL (apo, or TSEE mutant in 1:1 complex with RIPK3). **B** The brace helix suppresses the membrane permeabilization activity of 4HB domain in monomeric MLKL. **C** Evaluation of necroptotic signalling by wild-type and tetramerization interface mutants of full-length human MLKL in *MLKL*⁻/⁻ HT29 cells. Human MLKL expression (wild-type or mutant) was induced with doxycycline (Dox; 100 ng/mL) and cell death was measured in the presence or absence of necroptotic stimulus (TNF, Smac mimetic, IDN-6556; TSI) at 24 h. Cell death was quantified as a percentage of SYTOX Green positive cells using IncuCyte SX5 live cell imaging. Two independent cell lines were generated for wild-type and MLKL mutants. One independent line was assayed in *n* = 1 and the other in *n* = 2, for a total of *n* = 3

independent experiments. Data are plotted as mean ± SEM. **D** Cytosol (c) and membrane (m) fractions resolved by Blue Native-PAGE from transduced *MLKL*⁻/⁻ HT29 cells treated with doxycycline (Dox, 100 ng/mL) in the presence or absence of necroptotic stimulus (TSI; 4.5 h). Fractionation was verified by probing for VDAC (membrane) and GAPDH (cytosol) via SDS-PAGE. Images are representative of *n* = 2 independent experiments. **E** *MLKL*⁻/⁻ HT29 cells stably transduced with *MLKL*^WT^ or *MLKL*^R30E^ were treated with doxycycline (Dox) only or doxycycline plus necroptotic stimulus (Dox, TSI) in the presence or absence of MLKL inhibitor necrosulfonamide (NSA) or RIPK3 kinase inhibitor GSK872. IncuCyte SX5 imaging quantified the percentage of cell death (SYTOX Green-positive cells) every hour for 23 h. Data are plotted as mean ± SEM for *n* = 4 for wild-type MLKL and *n* = 6 for R30E MLKL. Two independent cell lines were generated for each of the wild-type and MLKL R30E mutant; WT cell lines were assayed in *n* = 2 and R30E mutant cell lines were assayed in *n* = 3 independent experiments.

residues within its sequence[47]. Our model identifies an extended brace helix that forms a coiled-coil, which is at the heart of this tetramer model and contributes an extensive array of hydrophobic side chains to the hydrophobic core of the coiled coil. This model explains why previously reported L162G/L165G mutant[47], along with I158E, M169E, L176D, L172D mutants reported herein, abolished MLKL oligomerization and necroptosis (Fig. 5b). Consistent with previous reports that MLKL forms disulfide-linked oligomers[47], we identified C184 in human MLKL as engaging in intermolecular disulfide bonding in the recombinant human MLKL tetramer. The close proximity of C184 between MLKL protomers in our MLKL tetramer model accounts for their propensity to form disulfides, and also provides further validation of this model, even though this disulfide bond is dispensable for necroptosis.

The flexibility of the loops connecting each 4HB domain to its neighbouring brace region limits certainty of the positions of 4HB domains in our negative stain electron micrographs. However, functional studies reported previously[40,59] and herein are highly consistent with the tetrameric arrangement of 4HB domains in our structural model (Fig. 5a), with a number of mutants that disrupt 4HB domain interfaces reported to impair necroptosis signalling[40,59]. Introducing charged repulsion in the core of the 4HB domain tetramer by Q117E and Q120E mutants impaired cell death in the present study. Similarly, MTSET alkylation of the recombinant Q120C human MLKL 4HB domain-brace protein was previously reported to inhibit membrane permeabilization[62]. Our structural model suggests that 4HB domain interfaces rely on several salt bridges between the α2 helix and the α4′ helix from the adjacent protomer (Fig. 3i). These and published data assert the importance of the α2 R34-α4′ D107 salt bridge; with disruption of this interaction abolishing necroptosis[40,49,59]. Previously, the 4HB domain α4 helix was identified as critical to facilitating MLKL translocation to the plasma membrane, but not to assembly of MLKL oligomers in cells[40,59]. As a result, it was proposed that this region of the 4HB domain may facilitate interaction with trafficking chaperones. While we cannot exclude this possibility, our data here raise the prospect that the disruption of the α2 helix R34-α4′ helix D107 salt bridge may perturb higher order assembly of the 4HB domains into a killer conformation. In such a scenario, Monobodies that bind to the human MLKL 4HB domain α4 helix, such as Mb33 and Mb37[59], would not be expected to prevent MLKL tetramerization, but could block the unleashing of the 4HB domain from the brace region and subsequent membrane engagement. This model supports the idea that 4HB domain reorganization within human MLKL tetramers is a prerequisite for plasma membrane association, where higher order MLKL hotspots capable of permeabilizing lipid bilayers assemble.

In all previously reported human 4HB domain structures, the N-terminus of the brace helix packs against the 4HB domain α2 and α4 helices, which was proposed to be an auto-inhibited conformation[43,49,63]. The executioner function of the 4HB domain

was proposed to be activated via a 'plug-release' mechanism[43], such that the brace region is released from the 4HB domain. Indeed, in our model of MLKL tetramer, release from the brace region enabled tetramerization of the 4HB domain, and formation of a positively charged cavity poised to interact with negatively-charged PIP headgroups. More excitingly, when an electrostatic repulsion was introduced to disrupt 4HB-brace interface by the R30E mutation, stable expression of mutant MLKL alone is sufficient to cause stimulus-independent cell death, which is inhibitable by RIPK3 inhibitor GSK'872, but not by necrosulfonamide. R30E MLKL's lack of susceptibility to NSA is consistent with an earlier report[49] but is nonetheless surprising. NSA is reported to function via a mechanism distinct from augmentation of the 4HB domain-brace helix latch to prevent 4HB domain release[63], and therefore overriding the latching mechanism with the R30E MLKL mutation would not be expected to impact NSA susceptibility. These findings raise the possibility that NSA may be unable to covalently modify Cys86 within the R30E MLKL mutant, and accordingly fails to disrupt R30E MLKL's association with cellular membranes. Intriguingly, constitutive cell death by MLKL R30E mutant does not require the phosphorylation of MLKL, indicating that GSK´872 may block cell death via abrogation of RIPK3 S227/T224 autophosphorylation, which is required for the recruitment of MLKL to RIPK3 or the necrosome[29,30,66].

Taken together, our study unifies the current structural and biochemical understanding of human MLKL activation by uncovering the interplay between the component domains. We propose that, during necroptosis, RIPK3-mediated phosphorylation of MLKL promotes not only release from the necrosome, but also MLKL tetramerization. In turn, higher order assembly of MLKL enables the unleashing of the 4HB executioner domain from an autoinhibited conformation to invoke MLKL's membrane association and executioner function.

## Methods

### Recombinant protein expression and purification

All recombinant human MLKL pseudokinase domain (residues 190–471) and full-length MLKL (residues 2–471) complexes with Monobodies or RIPK3 kinase domain (residues 2–316; C3S, C110A) were expressed in *Sf*21 insect cells using bacmids prepared in DH10MultiBac *E. coli* (ATG Biosynthetics) from pFastBac Htb vectors using established procedures[69]. Oligonucleotide sequences used in PCR amplifications for subcloning and mutagenesis are shown in Supplementary Table 2. Unphosphorylated human MLKL pseudokinase domain[32,42] and full-length wild-type human MLKL[40,41] were prepared using established methods. Briefly, all bacmids were introduced into $0.9 \times 10^6$ *Sf*21 cells (Merck) cultured in Insect-XPRESS (Lonza) media in 6-well plates using the Bac-to-Bac protocol (ThermoFisher Scientific). After 4 days of static incubation at 27 °C in a humidified incubator, the resulting P1 baculovirus was harvested and added to 25 mL *Sf*21 cells at $0.5 \times 10^6$ cells/mL density at 4% v/v, which were

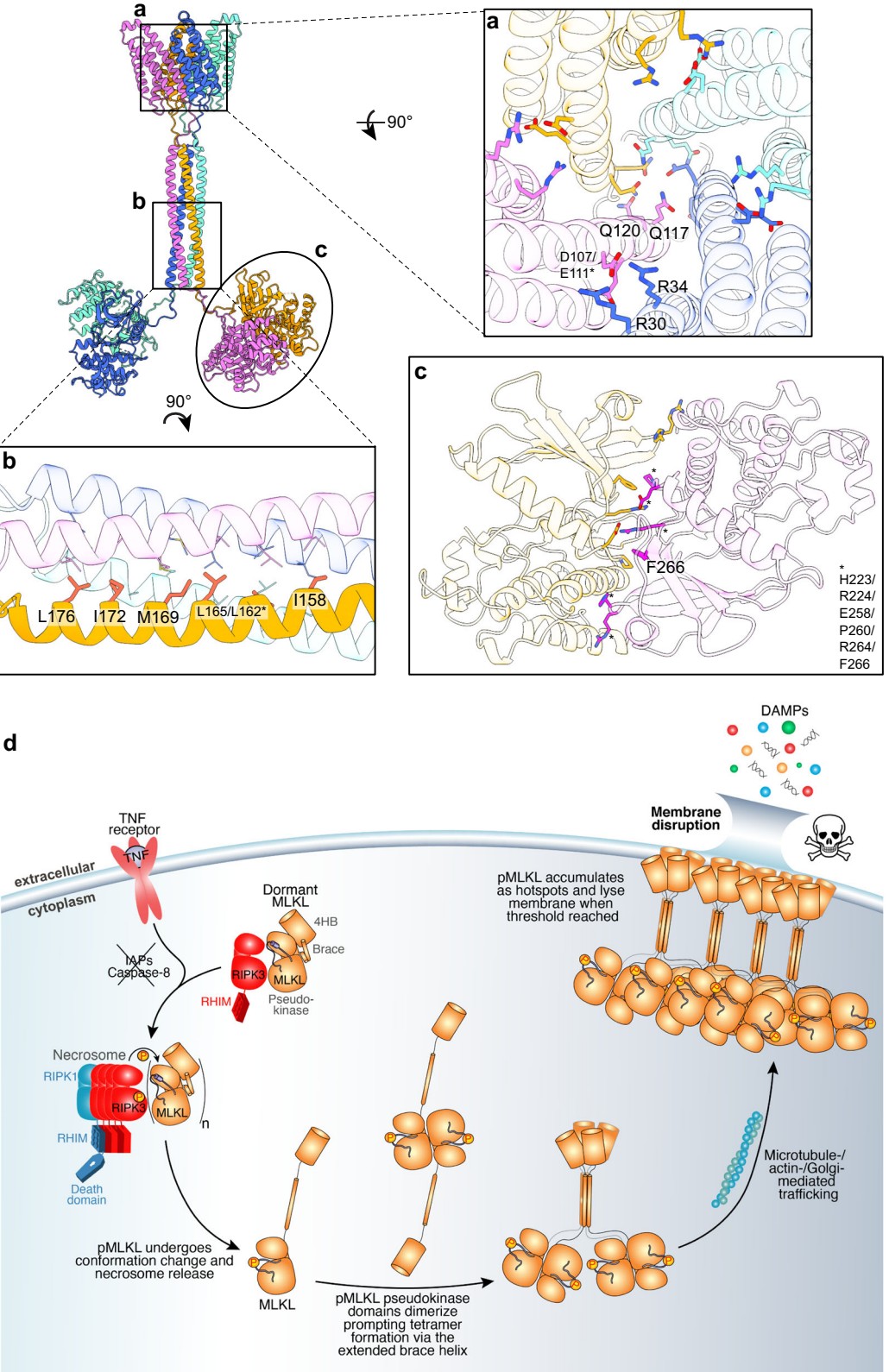

**Fig. 5 | Assembly of active MLKL tetramer requires interactions mediated by three domains. a–c** MLKL tetramerization interface residues (sticks) whose mutation disrupted necroptosis mapped on the model of full-length human MLKL tetramer (mutants reported by previous studies[40,47,52] are labelled with asterisks). **a** Four-helix bundle (4HB) domain tetramerization interface (top-down view); **b** brace helix coiled coil; **c** pseudokinase domain dimerization interface (zoomed panel shows the phosphorylated (p)-MLKL pseudokinase domain crystal structure, PDB: 8SLZ). **d** Proposed mechanism by which RIPK3-mediated phosphorylation drives MLKL tetramerization and cell death during necroptosis. The skull and crossbones image (Mycomorphbox_Deadly.png; by Sven Manguard) was used under a Creative Commons Attribution-Share Alike 4.0 license.

shaken at 27 °C, 130 rpm. The cell density was monitored using a haemocytometer slide and maintained at $0.5-3.0 \times 10^6$ cells/mL by diluting with fresh media when necessary until growth arrest (defined as a cell density less than the twice the cell count 1 day prior). Approximately 24 h after growth arrest was recorded, the P2 baculovirus was harvested by collecting the supernatant after pelleting the cells at $500 \times g$ for 5 min. P2 baculovirus was added to 0.5 L Sf21 cells cultured at $1-1.5 \times 10^6$ cells/mL in 2.8 L Fernbach flasks at 0.1% v/v, and cultured at 27 °C, 90 rpm, until 24 h post growth arrest. Cells were harvested at $500 \times g$ and pellets snap frozen in liquid $N_2$ and either thawed immediately for lysis or stored at −80 °C until required.

Phosphorylated human MLKL pseudokinase domain was generated by co-expressing human MLKL pseudokinase domain (residues 190-471) with human RIPK3 kinase domain (residues 1–316; harbouring C3S and C110A substitutions to prevent disulfide bond formation) in Sf21 insect cells. A previously reported co-expression pFastBac Htb vector[33,69] encoding a TEV protease-cleavable His$_6$ fusion N-terminal to the human MLKL pseudokinase domain (residues 190–471) under the polh promoter and a p10 promoter-human RIPK3 (residues 1–316; C3S, C110A)-HSV TK poly(A) signal (from pAceBac2; ATG Biosynthetics) cassette was further modified by introducing an N-terminal, HRV 3C protease cleavable GST-tag to the human RIPK3 gene by In-Fusion (Takara Bio) cloning into the position after the start codon. A bacmid generated from this vector was introduced into Sf21 cells, viruses were amplified and Sf21 cells were infected as detailed above. Sf21 cell pellets were lysed by sonication in Ni-NTA buffer (20 mM Tris pH 8.0, 500 mM NaCl, 20%v/v glycerol) supplemented with 10 mM imidazole (pH 8.0), EDTA-free cOmplete Protease Inhibitor (Roche), and 0.5 mM Bond-Breaker TCEP [tris-(2-carboxyethyl)phosphine] (ThermoFisher Scientifc), before the lysate was clarified by centrifugation ($40,000 \times g$, 30 min, 4 °C). The proteins were maintained at 4 °C throughout purification. The clarified lysate was incubated with Ni-NTA resin (Roche) pre-equilibrated in Ni-NTA buffer with 5 mM imidazole for 1 h on rollers at 4 °C, before the beads were pelleted at $500 \times g$ and washed extensively with Ni-NTA buffer containing 35 mM imidazole. The proteins were eluted in Ni-NTA buffer containing 250 mM imidazole. The eluate contains a mixture of His$_6$-p-MLKL pseudokinase domain dimer and His$_6$-(unphosphorylated) MLKL pseudokinase domain:GST-RIPK3 kinase domain heterodimer. These proteins were desalted into GST buffer (20 mM Tris pH 8.0, 200 mM NaCl, 10%v/v glycerol) using a 5 mL HiTrap Desalting column (Cytiva), before being supplemented with 1 mM ATP and 10 mM MgCl$_2$ to allow RIPK3 to phosphorylate MLKL, and TEV protease and TCEP (0.5 mM) to remove His$_6$-tag from MLKL. This mixture was incubated at 4 °C for ~18 h, before the TEV protease and uncleaved MLKL were removed by Ni-NTA pulldown and the GST-RIPK3 kinase domain was removed by GST pulldown (Glutathione Xpure Agarose resin, UBPBio). The eluate was filtered (0.45 μm) and spin concentrated (30 kDa MWCO; Millipore), before being loaded onto a HiLoad 16/160 Superdex 200 column (Cytiva) pre-equilibrated with SEC buffer [20 mM HEPES pH 7.5, 200 mM NaCl, 5% v/v glycerol]. After assessing purity by reducing StainFree SDS-PAGE (Bio-Rad), fractions containing purified p-MLKL were pooled and concentrated to 5 mg/mL, before being snap frozen in liquid nitrogen for storage at −80 °C.

For p-MLKL dephosphorylation studies and analytical SEC, recombinant p-MLKL pseudokinase domain (2 mg/mL) in SEC buffer supplemented with 2 mM MnCl$_2$ was mixed with recombinant lambda phosphatase (0.4 mg/mL) and incubated overnight at 4 °C in a final reaction volume of 50 μL. The next morning, the 50 μL reaction was injected onto a Superdex 200 10/300 Increase column (Cytiva) in SEC buffer at a flow rate of 0.5 mL/min. Gel filtration standard (BioRad, CA) was used to define the molecular weight marker elution volumes (1.35–670 kDa).

The expression constructs for full-length human MLKL in complex with RIPK3 or Monobodies were generated by modifying the previously reported pFastBac-derived vector encoding full-length human MLKL (residues 2–471; wild-type or T357E/S358E mutant) bearing an N-terminal, TEV-cleavable GST tag[40,41]. These co-expression vectors were generated by introducing an expression cassette [p10 promoter-protein of interest-HSV TK poly(A) signal] from a pAceBac2 vector (ATG Biosynthetics) encoding either human RIPK3 kinase domain (residues 1–316; C3S, C111A) or a Monobody into the non-coding region between BbsI and SnaBI sites of the pFastBac full-length human MLKL vector backbone. Bacmids generated from these vectors were introduced into Sf21 cells, viruses amplified and Sf21 cells infected as detailed above. Sf21 cell pellets were lysed by sonication in GST buffer supplemented with EDTA-free cOmplete Protease Inhibitor (Roche) and 0.5 mM Bond Breaker TCEP (ThermoFisher), before the lysate was clarified by centrifugation. The clarified lysate was incubated with Glutathione Xpure Agarose resin (UBPBio) pre-equilibrated in GST buffer for 1 h at 4 °C, before the beads were pelleted at $500 \times g$ and washed extensively with GST buffer. The proteins were eluted in GST buffer supplemented with 10 mM glutathione (pH 8.0). The eluate was supplemented with TEV protease and 0.5 mM TCEP and dialysed into SEC buffer at 4 °C overnight. TEV protease was removed from the dialysate by Ni-NTA pulldown, before being filtered (0.45 μm) and spin concentrated (30 kDa MWCO; Millipore) to load onto a HiLoad 16/160 Superdex 200 column (Cytiva) pre-equilibrated with SEC buffer. After assessing purity by reducing StainFree SDS-PAGE (Bio-Rad), fractions containing purified MLKL complexes were pooled and concentrated to 5 mg/mL, before being snap frozen in liquid $N_2$ for storage at −80 °C.

Recombinant Monobodies (in pPROEX Htb; Invitrogen) were generated from E. coli BL21-Codon Plus (DE3)-RIL using established procedures[33,59]. Briefly, transformed E. coli were grown as a starter culture in Super broth containing 100 μg/mL ampicillin (Roche) and used to inoculate 1 L Super broth containing 100 μg/mL ampicillin. Cultures were grown to OD$_{600}$ of ~0.6–0.8 at 37 °C, 200 rpm before the temperature was reduced to 18 °C and expression induced by addition of 0.5 mM IPTG overnight. Cells were pelleted at $4000 \times g$, lysed using Ni-NTA buffer by sonication, and debris removed by centrifugation at $40,000 \times g$; all steps were performed at 4 °C. The clarified and filtered (0.45 μm) lysate was loaded onto a HisTrap HP 5 mL column (Cytiva) equilibrated with Ni-NTA buffer supplemented with 10 mM imidazole (pH 8.0) at 4 °C. After washing in 20 mL of the same buffer, the bound protein was eluted over a linear gradient over 30 mL into Ni-NTA buffer containing 250 mM imidazole (pH 8.0) using an AKTA Pure instrument. The eluate was further purified by cleaving the His$_6$ tag using TEV protease (for applications where the absence of His$_6$ tag was desirable), dialysis, Ni-chromatography to eliminate uncleaved material and the TEV protease, and snap frozen for storage at −80 °C.

Recombinant human MLKL 4HB domain (residues 2-123) was expressed in E. coli BL21-Codon Plus (DE3)-RIL with an N-terminal TEV protease-cleavable GST tag by modifying a previously reported procedure[41]. Briefly, a HRV 3C cleavage site was introduced into the previously reported expression construct of human MLKL 4HB domain plus the first brace helix (residues 2–154) to replace the flexible loop between the 4HB domain and the brace region (123–130). Expression was performed in Super broth supplemented with 100 μg/mL Ampicillin as described above for His$_6$-Monobodies. Pelleted cells were resuspended in GST buffer (200 mM NaCl, 20 mM HEPES pH 7.5, 5% glycerol) supplemented with cOmplete protease inhibitor cocktail (Roche) and 0.5 mM Bond Breaker TCEP (Thermo), and lysed by sonication before debris was removed by centrifugation at $40,000 \times g$; all steps were performed at 4 °C. After allowing the GST-tagged proteins to bind to Glutathione Xpure Agarose resin, the resin was incubated with TEV protease and HRV 3C protease and 0.5 mM TCEP at 4 °C overnight. The unbound proteins were washed off in GST buffer, before the protein solution was supplemented with additional NaCl to a final concentration of 1 M to disrupt the electrostatic interaction between the 4HB domain and the cleaved brace region. The proteins in

high salt solution were concentrated before being loaded onto a Superdex 75 column (Cytiva) pre-equilibrated with high salt SEC buffer [20 mM HEPES (pH 7.5), 1 M NaCl, 5% v/v glycerol].

## Protein crystallization and structure determination

Recombinant p-MLKL pseudokinase domain (5 mg/mL in SEC buffer) was subjected to sitting drop crystallization trials conducted at the Monash Macromolecular Crystallization Facility (Clayton, VIC) with 100 nL protein solution mixed with 100 nL reservoir solution, and a 50 μL reservoir well volume. p-MLKL pseudokinase domain crystals grown in 0.1 M HEPES (pH 7.5), 20% w/v polyethylene glycol 8000 at 20 °C was soaked in cryoprotectant (reservoir solution supplemented with 20% v/v ethylene glycol) and flash-frozen in liquid nitrogen. The X-ray diffraction data were collected at the Australian Synchrotron MX2 beamline using the EIGER X 16M detector[70]. The diffraction data were indexed and integrated in XDS (Build 20161205)[71], and merged and scaled in AIMLESS (v0.7.8)[72]. Phases were solved by molecular replacement in PHASER 2.8.3[73] using the crystal structure of human MLKL pseudokinase domain (PDB: 4MWI) as the search model[42]. COOT (0.9 EL ccp4) and phenix.refine (1.18.2-3874-000) were used for iterative real-space and reciprocal-space refinement by manual model building and phase refinement[74,75]. Protein interfaces and the biological assembly in solution were analysed using PDBePISA[51]. Structure models were validated using MolProbity[76]. Structure figures were generated using either UCSF ChimeraX (1.2.5)[77] or PyMOL (v2.5.0; The PyMOL Molecular Graphics System, Schrödinger, LLC.). Structure alignments were calculated using UCSF ChimeraX. Data collection and processing statistics are summarized in Table 1 and illustrative images of the crystal lattice and density are shown in Supplementary Fig. 6.

## Small-angle X-ray scattering

SAXS data were collected using the inline-SEC setup at the Australian Synchrotron SAXS/WAXS beamline, including the implementation of co-flow to prevent exposure of the protein to the capillary wall[78,79]. MLKL pseudokinase domain:Monobody complexes were pre-purified by loading a mixture of equal amount of MLKL pseudokinase domain and Monobody onto a Superdex 75 10/300 column (Cytiva). For SEC-SAXS, the analytes were eluted from a Superdex 200 5/150 Increase column (Cytiva) in SEC buffer at a flow rate of 0.2 mL/min via a capillary in the path of the SAXS beam. Data were reduced, 2D radially integrated, and background scatter subtracted (using exposures from early in the SEC run) using Scatter-Brain (developed in-house by Stephen Mudie; V2.71). Exposures from the protein elution peak were averaged and subjected to analyses using the ATSAS 3.03 package[80] including Guinier analyses in PRIMUS, $P_r$ in GNOM, and comparisons with theoretical and experimental scatter using CRYSOL. Data collection and processing statistics are summarized in Table 2.

## Intact mass spectrometry with inline buffer exchange

To verify the phosphorylation status of recombinant p-MLKL pseudokinase domain, the purified proteins were separated by reverse-phase chromatography on a 25 cm ProSwift RP-4H monolith column (Thermo Scientific) using a micro-flow HPLC (M-class, Waters). The HPLC was coupled to an Impact II QTOF mass spectrometer (Bruker) equipped with an ESI source. Proteins were loaded directly onto the column for online buffer exchange at a constant flow rate of 30 μL/min with buffer A (99.9% Milli-Q water, 0.1% formic acid (FA)) using a trap-valve to direct flow to waste. After 10 min, the trap valve was switched to direct flow to the mass spectrometer, and proteins were eluted with a 15-min linear gradient from 2 to 90% buffer B (99.9% acetonitrile (ACN), 0.1% FA). The Impact II QTOF was operated in mode using Compass Hystar 5.1. Settings for the 40 samples per day method were as follows: Mass Range 100 to 3000 $m/z$, Capillary Voltage 4500 V, Dry Gas 4 L/min, Dry Temp 200 °C.

## Tryptic digestion and mass spectrometry analysis

Recombinant MLKL protein was digested in solution. Protein was denatured in 6 M Urea in 50 mM ammonium bicarbonate, and alkylated using 10 mM iodoacetamide for 30 min in the dark. The solution was then diluted in 50 mM ammonium bicarbonate to 1 M Urea before being digested with trypsin overnight at 37 °C. Tryptic peptides were desalted using C18 StageTips (2× plugs of 3 M Empore resin, 2215[81]) and lyophilized to dryness using CentriVap (Labconco) prior to reconstituting in 20 μL of 0.1% FA/2% ACN ready for mass spectrometry analysis. Peptides (5 μL) were separated by reverse-phase chromatography on a $C_{18}$ fused silica column (inner diameter 75 μm, OD 360 μm × 25 cm length, 1.6 μm $C_{18}$ beads) packed into an emitter tip (IonOpticks), using a Dionex 3000 RSLC nano-liquid chromatography system (ThermoFisher) and PAL RSI sample robot (CTC) with custom valve (in-house). The HPLC was coupled to a timsTOF Pro (Bruker) equipped with a CaptiveSpray source. Peptides were loaded directly onto the column at a constant flow rate of 400 nL/min with buffer A (99.9% Milli-Q water, 0.1% FA) and eluted with a 30-min linear gradient from 2 to 34% buffer B (99.9% ACN, 0.1% FA). The timsTOF Pro was operated in PASEF mode using Compass Hystar 5.1. Settings for the 30 samples per day method were as follows: Mass Range 100–1700 $m/z$, 1/K0 Start 0.6 V·s/cm$^2$ End 1.6 V·s/cm$^2$, Ramp time 110.1 ms, Lock Duty Cycle to 100%, Capillary Voltage 1600V, Dry Gas 3 L/min, Dry Temp 180 °C, PASEF settings: 10 MS/MS scans (total cycle time 1.27 sec), charge range 0–5, active exclusion for 0.4 min, Scheduling Target intensity 10,000, Intensity threshold 2500, CID collision energy 42 eV. Raw files were analysed using MaxQuant (version 1.6.17.0)[82]. The database search was performed using a Uniprot *Homo sapiens* database plus common contaminants with strict trypsin specificity allowing up to two missed cleavages. MaxQuant APL files were converted to MGF files using the APL to MGF convertor software (https://www.wehi.edu.au/people/andrew-webb/1298/apl-mgf-converter). Crosslinked peptides were identified from the MGF files using the MeroX software (version 2.0.1.4)[83]. Cysteine residues were set as specificity sites for disulfide crosslinking (-H2). Trypsin was set as the enzyme allowing for three missed cleavages. Maximum peptide length was set to 40. Precursor precision was set at 10 ppm with fragment ion precision set at 20 ppm. Spectra associated with the Cys184–Cys184' disulfide bonds are presented in Supplementary Fig. 3a.

## Phosphoproteomic analysis

Raw files consisting of high-resolution MS/MS spectra were processed with MaxQuant (version 1.6.17.0) for feature detection and protein identification using the Andromeda search engine as previously described[84]. Extracted peak lists were searched against the reviewed Homo sapiens (UniProt, August 2020) database as well as a separate reverse decoy database to empirically assess the false discovery rate (FDR) using strict trypsin specificity, allowing up to 2 missed cleavages. The mass tolerance for precursor ions and fragment ions were 20 ppm and 0.5 Da, respectively. The search included fixed modification of carbamidomethyl (cysteine) and variable modifications including oxidation (methionine), phosphorylation (serine, threonine or tyrosine) and Protein-N terminal acetylation. PSM and protein identifications were filtered using a target-decoy approach at an FDR of 1%. MS spectra were manually validated using IPSA (http://www.interactivepeptidespectralannotator.com)[85].

## Ni-NTA co-precipitation assay

Recombinant full-length human MLKL (wild-type or T357E/S358E mutant) co-expressed with RIPK3 (20 μg) was mixed with each Monobody (20 μg) in 100 μL Ni-NTA buffer supplemented with 10 mM imidazole (pH 8.0) at 4 °C, before 2 μL of this mixture was removed and mixed with 18 μL Ni-NTA buffer as the input sample to be analysed by SDS-PAGE. Each protein mixture was mixed with 40 μL 50% Ni-NTA resin slurry pre-equilibrated in the same buffer and incubated for

30 min at 4 °C with gentle agitation. The Ni-NTA resin was pelleted by centrifugation (500 × g) and washed extensively with the same buffer. The bound proteins were eluted twice with 50 µL Ni-NTA buffer containing 250 mM imidazole (pH 8.0). The eluates were diluted fivefold with Ni-NTA buffer, before the input samples and eluates were analysed by reducing StainFree SDS-PAGE (Bio-Rad).

## Computational modelling of human MLKL tetramer

The sequence of full-length human MLKL (Uniprot Q8NB16) was truncated into two sections (residues 1–179 and residues 147–471) overlapping via the C-terminal region of the brace helix (residues 147–179). ColabFold-Multimer[60] (version 2) was used to generate a tetrameric prediction for each sequence. Five models were generated for the N-terminal segment. Of the five, only one (ranked highest by pLDDT score) had a convincing tetrameric interface with large buried surface area. This model was selected based on its ColabFold ranking and its agreement with Cai et al.'s mutational data[47]. Due to computational limits, only two models were generated for the larger C-terminal segment. The two models were highly similar; the brace region coiled coil and the pseudokinase dimer sections were almost identical and the models only varied in the relative orientation of these two sections to one another. The model with the better pLDDT score was selected for further modelling. The two models were aligned in PyMOL using their overlapping region and merged into one model (four chains of full-length MLKL) in COOT. We note that upon repeating structure predictions for the N-terminal segment using ColabFold-Multimer version 3 different structures are generated; this version favours a 4HB domain conformer with a latched brace region model.

The p-MLKL pseudokinase domain dimer crystal structure was linked to the MLKL tetramer model from ColabFold by first structurally aligning the dimeric crystal structure to the pseudokinase dimers in each chain, using the CEAlign algorithm[86] in PyMOL 2.0. After structural alignment, the pseudokinase domain dimers in the predicted model were deleted and dimeric crystal structures were linked to the brace region using Rosetta (v3.5) next-generation kinematic loop modelling[87]. 100 structures for each chain were produced and scored using the REF2015 scoring model[88]. The lowest energy structures from each chain were combined into one complex. Energy minimization was then performed on the entire complex using the Rosetta Fast Relax protocol[61]. Coordinate constraints were added to the backbone heavy atoms to avoid large conformational changes during relaxation, focusing instead on reduction of atomic clashes and improvement of torsion angles. To evaluate the quality of the final structure, Ramachandran analysis was performed using MolProbity (4.02b-467) to assess optimal ψ and φ angles[76].

## Negative stain electron microscopy

Purified proteins were diluted in buffer (20 mM HEPES pH 7.5, 200 mM NaCl) to appropriate concentrations (10 µg/mL for full-length human MLKL tetramer; 8 µg/mL for full-length MLKL in complex with Mb27) and applied to glow discharged 400 mesh copper grids with carbon support films (Ted Pella Inc., Prod No. 01844-F). The grids were blotted (and washed in MilliQ water for MLKL in complex with Mb27) and stained with uranyl acetate (1% w/v for MLKL tetramer, 2% w/v for MLKL in complex with Mb27). Data were collected using a Talos L120C transmission electron microscope (ThermoFisher Scientific; Ian Holmes Imaging Facility, Bio21 Institute of Molecular Science and Technology, Parkville, Australia) at 120 kV, with a pixel size of 2.42 Å/pixel at the specimen level for MLKL tetramer, and 1.89 Å/pixel for MLKL in complex with Mb27. Images were recorded with a Ceta camera (ThermoFisher Scientific) at varying defocus (−0.25 to −0.6 µm) and processed with Relion (v3.1.3). Particles were auto-picked with Laplacian of Gaussian filters (200–380 Å for MLKL tetramer; 250–380 Å for MLKL in complex with Mb27) and extracted (box size of 168 pixels for MLKL tetramer, 214 pixels for MLKL in complex with Mb27), before iterative 2D classification (mask diameter of 350 Å for MLKL tetramer, 380 Å for MLKL in complex with Mb27) to remove junk particles.

## Generation of cell lines

DNA sequences encoding human MLKL mutants were synthesized and subcloned into the pFTRE3G PGK puro vector[32] by ATUM (CA). The C184S human MLKL mutant construct was prepared in-house by overlap mutagenesis PCR (oligonucleotide sequences in Supplementary Table 2), and subcloned as a BamHI-EcoRI fragment into the pFTRE3G PGK puro vector. Midiprep DNA was co-transfected into HEK293T cells (originally sourced from ATCC) with helper plasmids pVSVg and pCMV δR8.2 to generate lentiviral particles using Effectene (Qiagen). $MLKL^{-/-}$ HT29 cells[40] derived from HT29 cells originally sourced from ATCC were then stably transduced with the resulting lentivirus and successful transductants selected using puromycin (1.25 µg/mL; StemCell Technologies) as before[32,38]. Transduced $MLKL^{-/-}$ HT29 cells were cultured in DMEM + 8% FCS, at 37 °C and 10% (v/v) $CO_2$. Cells were not formally authenticated, but their morphologies and responses to stimuli were consistent with their designations. All mammalian cell lines were routinely PCR monitored for mycoplasma contamination.

## Reagents and antibodies

Antibodies: rat anti-human MLKL 7G2 (1:1000–2000; produced in-house[89]; available from Merck-Millipore as MABC1636) was produced in-house. Rabbit anti-VDAC (AB10526; 1:10,000) was purchased from Millipore, rabbit anti-GAPDH (#2118; 1:5000) was purchased from Cell Signalling Technology, mouse anti-actin (A1978; 1:5000) was purchased from Sigma Aldrich, mouse anti-FLAG M2 (F1804; 1:3000) was purchased from Sigma Aldrich, and rabbit anti-human pMLKL (EPR9514; 1:1000–3000) was purchased from Abcam. Cell agonists/antagonists: recombinant human TNF-Fc (100 ng/mL; produced in-house[90]), Smac mimetic Compound A (500 nM; Tetralogic Pharmaceuticals as in ref. 27), Pan-caspase inhibitor IDN-6556 (5 µM; Tetralogic Pharmaceuticals), necrosulfonamide (2 µM; Merck #480073) and GSK´872 (10 µM; SynKinase #SYN-5481).

## Western blot

$MLKL^{-/-}$ HT29 cells were seeded into 48-well plates at 60,000 cells/well and left to settle for 6 h. Cells were then induced overnight with doxycycline (100 ng/mL) to stimulate MLKL expression. Following doxycycline pretreatment, cells were stimulated with TNF (100 ng/mL), Smac mimetic Compound A (500 nM), and Pan-caspase inhibitor IDN-6556 (5 µM) for 4 h. Cells were harvested in 2× SDS Laemmli's lysis buffer, boiled at 100 °C for 10–15 min, and then resolved by 4–15% Tris-Glycine SDS-PAGE (Bio-Rad). Proteins were transferred to PVDF membrane and probed with the antibodies indicated.

## IncuCyte cell death assays

$MLKL^{-/-}$ HT29 cells were plated at 30,000 cells per well in a 48-well plate and left to settle for 24 h. HT29 cells were then pre-stimulated with doxycycline (100 ng/mL) overnight and where applicable in combination with inhibitors, necrosulfonamide (NSA; 2 µM) or GSK´872 (10 µM). In Phenol Red-free media supplemented with 1% FCS, 1 mM Na pyruvate, 1 mM L-GlutaMAX, SYTOX Green (Invitrogen S7020; 1:10,000), and DRAQ5 (ThermoFisher #62251; 1:5000), cells were stimulated with the outlined combination treatments of doxycycline (100 ng/mL), TNF (100 ng/mL), Smac mimetic Compound A (500 nM), Pan-caspase inhibitor IDN-6556 (5 µM), necrosulfonamide (NSA; 2 µM) or GSK´872 (10 µM). Images were taken every hour using IncuCyte SX5 imaging (v2022B or v2021B software) and cell death percentage values was quantified by number of SYTOX green positive (dead) cells out of total DRAQ5 positive (live) cell numbers.

## BN-PAGE

*MLKL*[-/-] HT29 cells were seeded into 6-well plates ($0.6 \times 10^6$ cells per well) and left to settle overnight. The following day, cells were treated with doxycycline (100 ng/mL) overnight to induce the expression of mutant MLKL constructs. Cells were then treated with necroptotic stimulants TNF (100 ng/mL), Smac mimetic Compound A (500 nM), and Pan-caspase inhibitor IDN-6556 ($5 \mu M$) for 4.5 h. Cells were harvested and permeabilised in MELB buffer (20 mM HEPES pH 7.5, 100 mM KCl, 100 mM sucrose, 2.5 mM MgCl₂, $2 \mu M$ *N*-ethyl maleimide, 0.25% v/v digitonin and protease and phosphatase inhibitors). To fractionate cells into cytoplasmic and crude membrane fractions, permeabilized cells were centrifuged (5 min, $11,000 \times g$) and fractions were further solubilized in 1% digitonin. Samples were resolved on a 4–16% NativePAGE (Invitrogen) gel and proteins transferred to PVDF membrane. Membranes were destained (50% v/v methanol, 25% v/v acetic acid), denatured (6 M guanidine hydrochloride, 5 mM β-mercaptoethanol, 10 mM Tris pH 6.8), blocked in 5% skim milk and then probed with indicated antibodies. Samples of each cytoplasm and crude membrane fraction were reduced in 5× SDS Laemmli's lysis buffer and then immunoblotted as per instructions in 'Western Blot' methods.

## Immunoprecipitation for BN-PAGE

HT29 cells stably transduced with doxycycline-inducible Mb27 and Mb32 constructs (bearing an N-terminal FLAG tag and a C-terminal GFP tag)[33] were seeded into 15 cm plates at $10 \times 10^6$ cells/plate and induced overnight with 500 ng/mL doxycycline, then treated with TNF (100 ng/mL), Smac mimetic (Compound A; 500 nM) and pan-caspase inhibitor IDN-6556 ($5 \mu M$) (TSI) for 4.5 h. Cells were harvested in FLAG wash buffer (50 mM Tris pH 7.4, 150 mM NaCl) supplemented with 0.5 mM TCEP, 1 mM EDTA, 2 mM sodium vanadate, 10 mM sodium fluoride, 1 mM PMSF, and cOmplete protease inhibitor tablet (Roche), before lysis by sonication on ice. The lysates were clarified by centrifugation at $20,000 \times g$ for 10 min at 4 °C. The supernatants were incubated with anti-FLAG M2 Affinity Gel (Millipore) for 1 h. Beads were washed in FLAG wash buffer before the proteins were eluted with FLAG wash buffer supplemented with 0.5 mg/mL FLAG peptide. The FLAG eluates were kept on ice until analysis by BN-PAGE.

## Reporting summary

Further information on research design is available in the Nature Portfolio Reporting Summary linked to this article.

## Data availability

Source data are published as an accompanying file; any additional data, including expression construct sequences, are available from the corresponding authors upon request. Any materials are available from the corresponding authors under Materials Transfer Agreement. The atomic coordinates for the crystal structure reported here have been deposited in the Protein Data Bank (PDB) with the accession code 8SLZ [https://doi.org/10.2210/pdb8SLZ/pdb] (p-MLKL pseudokinase domain). Previously reported crystal structures depicted in this work are available from the Protein Data Bank with accession codes: 7MON [https://doi.org/10.2210/pdb7MON/pdb], 7JXU [https://doi.org/10.2210/pdb7JXU/pdb], 7JW7 [https://doi.org/10.2210/pdb7JW7/pdb], 6UX8 [https://doi.org/10.2210/pdb6UX8/pdb], with 4MWI [https://doi.org/10.2210/pdb4MWI/pdb] used for phasing by molecular replacement. Source data are provided with this paper.

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

## Acknowledgements

Electron microscopy was conducted at the Ian Holmes Imaging Center. We thank the Monash Macromolecular Crystallization Center for assistance with screening for protein crystallization conditions, and the Australian Synchrotron MX and SAXS/WAXS beamline staff for assistance with data collection. This research was undertaken using the MX2 crystallography beamline at the Australian Synchrotron, Victoria, Australia and made use of the ACRF Detector. We acknowledge scholarship support for Y.M. (Melbourne Research Scholarship; AINSE PGRA scholarship), S.E.G. (Australian Government Research Training Program Stipend Scholarship; Wendy Dowsett Scholarship) and H.H. (Australian Government Research Training Program Stipend Scholarship). We are grateful to the National Health and Medical Research Council for fellowship (P.E.C., 2009062; J.M.M., 1172929), grant (J.M.H., 2011584) and infrastructure (IRIISS 9000719) support; and the Victorian Government Operational Infrastructure Support scheme. N.K. acknowledges funding support from the National Institutes of Health (R35 GM139656); A.G. is a CSL Centenary Fellow.

## Author contributions

Y.M. and S.E.G. designed, performed and analysed experiments; Y.M. wrote the paper with J.M.M.; K.A.D. designed and analysed experiments and structural models; Y.M., K.A.B., A.P.L., H.H. and A.G. performed electron microscopic data acquisition and analysis; C.R.H., J.M.H., C.F., S.N.Y., L.F.D. and T.D. performed and analysed experiments; A.V. and N.K. performed computational modelling; A.K. and S.K. identified the Monobodies used in this study; P.E.C. and J.M.M. supervised the project, analysed and interpreted data; all authors commented on the manuscript.

## Competing interests

P.E.C. and J.M.M. have received research funding from Anaxis Pharma Pty Ltd. The other authors declare no competing interests.
