## [Peer Review File · Nature Communications]

Phosphorylation-dependent pseudokinase domain dimerization drives full-length MLKL oligomerizationReviewers' Comments:

Reviewer #1:

Remarks to the Author:

The manuscript from Meng and team is a very well written piece of forefront research in the MLKL/necroptosis field, which employs structural biophysics, liposome permeabilization assays and cellular pathway analysis (nearly all in the context of the full-length human pseudokinase complex) to understand communication between the three component domains in MLKL, and how these are regulated to control necroptosis. The use of 2 monobodies and thorough and dynamic work to piece together the full length protein structure using negative stain EM and the ColabFold-Multimer software adds to the new mechanistic insights, and provides testable hypotheses that are taken forward; I particularly like the idea that the 4HB tetrameric bundle from the model can form a large basic pocket that could accommodate phosphoinositols (though this is not tested in the current work). The work has several highlights and essentially no weaknesses and it is important because of regulatory differences between human and murine MLKL. It does a great job of uniting a variety of published and novel in crystallo data with an array of credible solution and cellular data. In the context of impact, the druggability of MLKL is likely to become ever more important in a variety of disease indications, so a solid mechanistic platform such as this becomes ever more important to publish. Major findings include the demonstration that phosphorylation (by RIPK3) induces dimerization, which then leads to the assembly of MLKL into a membrane-disrupting tetramer. This would generate at least one rate-limiting step in the process, and the data are entirely consistent with a phosphorylation-triggered dimer to tetramer, and notably an 'elongated' 4HB tetramer in human MLKL, being necessary and sufficient to trigger membrane insertion and necroptosis. In particular, the MLKL R30E data are very convincing to this reviewer.

Minor comments

1) Line 108. Phosphorylation of MLKL. In Supp Fig 1a/1b, formation of a dimer in the presence of RIPK3, and intact MS-based MLKL analysis was employed. Does a phosphatase-treated dimer/tetramer collapse to a monomer by gel filtration? Also, can the '2 to 3' phosphorylation sites at least attempt to be formally mapped to T357/S358 acceptors using fragmentation of proteolysed MLKL, since these modifications can't formally be confirmed in the crystal structure, due to flexibility of the activation segment.

2) Line 206. The line states 'to prevent their phosphorylated'. This doesn't make sense to me, and should probably be 'to mimic constitutive phosphorylation...'

3) The C184 residues and their role in the MLKL tetramer are explored in the cellular context. Surprisingly, no effect of mutation was found on necroptosis. How well-conserved are these Cys residues in other MLKL species?

Signed

Patrick Eyers

Reviewer #2:

Remarks to the Author:

Meng et al. describe a mechanistic model for how phosphorylation of the pseudokinase MLKL by RIPK3 results in MLKL activation, oligomerization and membrane permeabilization as a key step in necroptosis. They present structural evidence for the effects of the phosphorylation of MLKL, biochemical validation of dimerization and tetramerization, and construct a structural model for the fully assembled active MLKL tetramer through a combination of computational modeling and negative stain electron microscopy. Finally, they present extensive functional data validating the molecular model and importance of specific protein-protein interfaces to MLKL function. The paper is interesting

and insightful, and the work is carried out in a nice methodical manner, transitioning smoothly from establishing biochemical mechanism at the level of individual domains to doing so in the full-length molecule. Altogether the paper makes a significant advance in our understanding of the molecular mechanisms underlying necroptosis. The manuscript is also well written and lucid, and the figures are particularly nice. The only substantial comment I have is that the validation of the hybrid computational-experimental tetramer model by electron microscopy is a little weak, and this part of the paper would benefit from some additional analysis (see below). Other than that I have only some minor points that the authors should consider which can be addressed by editing of the manuscript.

The validation of the full tetrameric model of MLKL by electron microscopy is a little weak as presented. It is mentioned that cryoEM analysis was problematic because of aggregation at the air-water interface, but negative stain should be capable of providing sufficient information to validate the gross features of the model. However, it is not clear that the 2D class averages shown really support the tetrameric model as claimed. Did the authors attempt a 3D reconstruction and docking of the molecular model into the EM density? This would at least more convincingly demonstrate consistency than just saying they are a similar size. The number of particles is a bit low and the 2D averages are missing half the structure; if conformational heterogeneity is an issue as suggested then collecting a larger number of particles should help (e.g. 20-30K) along with more detailed classification to resolve different conformations. Another option would be for the authors to collect data with and without the Mb27 monobody to demonstrate that the bilobal structures observed in the 2D class averages are indeed the pseudokinase domain dimers. Overall, some effort should be taken to tighten up the analysis presented in this section.

Minor comments:

Line:

50 - The authors might consider the use of the full phrase "pathogen associated molecular patterns" or PAMPs.

89 - "used" should be "using"

136 - Supp Fig. 1C is referenced to show the glu-lys salt bridge but this is not actually shown in the figure.

152 - It only becomes clear at this point in the manuscript that there are several published human MLKL structures in the active state. It might help to clarify what is unique about the author's own structure and exactly what it demonstrates that the other structures do not, i.e. role of phosphorylation specifically.

171 - missing a "the" in "the sizes of ... MLKL"

206 - typo, should be phosphorylation.

206 - So the phosphomimetic mutation doesn't trigger a switch to the activated dimeric form then? Perhaps the authors could clarify. Is this just an example of the failure of phosphomimetic mutations to really capture the effects of phosphorylation?

229 - tetramerization is misspelled

Figure 1fg: it might be nice to have the two scattering curves/fits compared on the same graph. As it is it is very hard to tell by eye that f matches the tetramer and g matches the dimer.

Figure 4c: lacking a y-axis title.

We thank the reviewers for their kind appraisal of our manuscript and their constructive suggestions for revision. We have now edited our text to incorporate each of the reviewers' suggestions, performed the three experiments they suggested (mass spectrometry and dephosphorylation-SEC analysis now included in Supplementary Figure 1, and improved negative-stain 2D images of MLKL tetramers without Mb27 now included as Figure 3c). Below we respond to each reviewer query (shown in *blue italics*) in turn with our comments (in plain black text), with the changes implemented in the manuscript highlighted in grey.

REVIEWER COMMENTS

Reviewer #1 (Remarks to the Author):

The manuscript from Meng and team is a very well written piece of forefront research in the MLKL/necroptosis field, which employs structural biophysics, liposome permeabilization assays and cellular pathway analysis (nearly all in the context of the full-length human pseudokinase complex) to understand communication between the three component domains in MLKL, and how these are regulated to control necroptosis. The use of 2 monobodies and thorough and dynamic work to piece together the full length protein structure using negative stain EM and the colabFold-Multimer software adds to the new mechanistic insights, and provides testable hypotheses that are taken forward; I particularly like the idea that the 4HB tetrameric bundle from the model can form a large basic pocket that could accommodate phosphoinositols (though this is not tested in the current work). The work has several highlights and essentially no weaknesses and it is important because of regulatory differences between human and murine MLKL. It does a great job of uniting a variety of published and novel in crystallo data with an array of credible solution and cellular data. In the context of impact, the druggability of MLKL is likely to become ever more important in a variety of disease indications, so a solid mechanistic platform such as this becomes ever more important to publish. Major findings include the demonstration that phosphorylation (by RIPK3) induces dimerization, which then leads to the assembly of MLKL into a membrane-disrupting tetramer. This would generate at least one rate-limiting step in the process, and the data are entirely consistent with a phosphorylation-triggered dimer to tetramer, and notably an 'elongated' 4HB tetramer in human MLKL, being necessary and sufficient to trigger membrane insertion and necroptosis. In particular, the MLKL R30E data are very convincing to this reviewer.

We are delighted with the positivity of the reviewer and thank them for their generous appraisal of this study.

Minor comments

1) Line 108. Phosphorylation of MLKL. In Supp Fig 1a/1b, formation of a dimer in the presence of RIPK3, and intact MS-based MLKL analysis was employed. Does a phosphatase-treated dimer/tetramer collapse to a monomer by gel filtration?

We thank the reviewer for their excellent suggestions to better understand the function of the phosphorylation event. We have now included a SEC trace in Supplementary Figure 1f to show the elution profile of the lambda phosphatase treated phospho-MLKL pseudokinase domain. We had initially predicted that once phosphorylated and locked into the dimer conformation that the MLKL pseudokinase domain would remain dimeric, however, this was not what we found in the experiment. PhosphoMLKL collapsed mostly to the monomeric form, which we now comment on in the main text (page 5, lines 117-120). This was an interesting observation, since it underscores the importance of phosphorylation or the role of Mb27 in stabilising the closed form of the pseudokinase domain, supporting the idea that this is the dimerization-competent conformer; we thank the reviewer for prompting further examination.

Also, can the '2 to 3' phosphorylation sites at least attempt to be formally mapped to T357/S358 acceptors using fragmentation of proteolysed MLKL, since these modifications can't formally be

confirmed in the crystal structure, due to flexibility of the activation segment.

We agree this is useful information and have now included these data as Supplementary Figure 1c-e of the revised version, and updated methods accordingly.

2) Line 206. The line states 'to prevent their phosphorylated'. This doesn't makes sense to me, and should probably be 'to mimic constitutive phosphorylation...'

Line 206 has been fixed to clarify 'to prevent their phosphorylation'.

3) The C184 residues and their role in the MLKL tetramer are explored in the cellular context. Surprisingly, no effect of mutation was found on necroptosis. How well-conserved are these Cys residues in other MLKL species?

We thank the reviewer for prompting elaboration on this point. We now include additional text on page 8 (line 213-216) to describe that this Cys is well conserved among primate and bird MLKL, but not in other species, such as rodents. Similar observations that Cys mutants (albeit at other sites) compromised mouse MLKL oligomerization, but not death signaling, have been reported; we now mention this report on page 13 (lines 400-402) in light of the reviewers' comment about the lack of effect of C184S human MLKL mutation being surprising.

*Signed
Patrick Eyers*

Reviewer #2 (Remarks to the Author):

Meng et al. describe a mechanistic model for how phosphorylation of the pseudokinase MLKL by RIPK3 results in MLKL activation, oligomerization and membrane permeabilization as a key step in necroptosis. They present structural evidence for the effects of the phosphorylation of MLKL, biochemical validation of dimerization and tetramerization, and construct a structural model for the fully assembled active MLKL tetramer through a combination of computational modeling and negative stain electron microscopy. Finally, they present extensive functional data validating the molecular model and importance of specific protein-protein interfaces to MLKL function. The paper is interesting and insightful, and the work is carried out in a nice methodical manner, transitioning smoothly from establishing biochemical mechanism at the level of individual domains to doing so in the full-length molecule. Altogether the paper makes a significant advance in our understanding of the molecular mechanisms underlying necroptosis. The manuscript is also well written and lucid, and the figures are particularly nice. The only substantial comment I have is that the validation of the hybrid computational-experimental tetramer model by electron microscopy is a little weak, and this part of the paper would benefit from some additional analysis (see below). Other than that I have only some minor points that the authors should consider which can be addressed by editing of the manuscript.

The validation of the full tetrameric model of MLKL by electron microscopy is a little weak as presented. It is mentioned that cryoEM analysis was problematic because of aggregation at the air-water interface, but negative stain should be capable of providing sufficient information to validate the gross features of the model. However, it is not clear that the 2D class averages shown really support the tetrameric model as claimed. Did the authors attempt a 3D reconstruction and docking of the molecular model into the EM density? This would at least more convincingly demonstrate consistency than just saying they are a similar size. The number of particles is a bit low and the 2D averages are missing half the structure; if conformational heterogeneity is an issue as suggested then collecting a larger number of particles should help (e.g. 20-30K) along with more detailed

classification to resolve different conformations. Another option would be for the authors to collect data with and without the Mb27 monobody to demonstrate that the bilobal structures observed in the 2D class averages are indeed the pseudokinase domain dimers. Overall, some effort should be taken to tighten up the analysis presented in this section.

We thank the reviewer for the excellent suggestions. We have now collected >120 micrographs on MLKL tetramer without Mb27, from which we were able to pick 30,000 particles. A larger number of particles indeed improved the quality of the 2D classes. We have now updated the Figure 3c with these data and updated the text to reflect this new approach (page 10, lines 287-289; and methods). The MLKL:Mb27 2D classes presented in our initial submission have been moved to Supp. Fig. 4a and are now mentioned on page 10 (lines 297-299).

Regarding 3D reconstruction, the conformational heterogeneity of the N-terminal 4HB domain remains a major challenge. As such, after iterative rounds of 2D classification to remove junk particles and different conformations, too many particles had to be removed to allow generation of a 3D reconstruction of meaningful quality. We expect that stabilization of the relative motion between each component domain, such as by using antibodies, may be required to reduce the conformational heterogeneity and fully resolve this structure. This remains the focus of ongoing work.

Minor comments:

Line:

50 - The authors might consider the use of the full phrase “pathogen associated molecular patterns” or PAMPs.

Line 51 has now been updated to use the full phrase “pathogen associated molecular patterns”

89 - “used” should be “using”

Line 90 has been fixed to “using”

136 - Supp Fig. 1C is referenced to show the glu-lys salt bridge but this is not actually shown in the figure.

Lines 143-145 have now been updated to correctly reference both Supplementary Figure 2a and Figure 1a, which correctly depict the E-K salt bridge.

152 - It only becomes clear at this point in the manuscript that there are several published human MLKL structures in the active state. It might help to clarify what is unique about the author’s own structure and exactly what it demonstrates that the other structures do not, i.e. role of phosphorylation specifically.

We thank the reviewer for prompting clarification on this important point. We have added text on page 6 of the revised manuscript (lines 146-149) to ensure that it is clear to the reader that the closed conformation has been observed in earlier MLKL pseudokinase domain structures, but there has been no concrete evidence to connect phosphorylation to driving the closed form. As a result, our structure provides important validation for the back-to-back, head-to-tail dimer seen in the crystal lattice as an arrangement driven by RIPK3-mediated activation of MLKL.

171 - missing a “the” in “the sizes of ... MLKL”

Line 183 has been fixed, with thanks.

206 - typo, should be phosphorylation.

Line 221 typo has been fixed, again with thanks to the reviewer for their careful reading of our manuscript.

206 - So the phosphomimetic mutation doesn't trigger a switch to the activated dimeric form then? Perhaps the authors could clarify. Is this just an example of the failure of phosphomimetic mutations to really capture the effects of phosphorylation?

This is an excellent question. In our hands, we do not see higher order assemblies of phosphomimetic full length MLKL when co-expressed with RIPK3 kinase domain, which leads us to conclude, as the reviewer suggested, that phosphomimetic mutations incompletely capture the effects of phosphorylation. We have now added a comment to the main text on page 8 to reflect the reviewer's important point.

229 - tetramerization is misspelled

Line 246 typo has now been fixed, with thanks.

Figure 1fg: it might be nice to have the two scattering curves/fits compared on the same graph. As it is it is very hard to tell by eye that f matches the tetramer and g matches the dimer.

Figure 4c: lacking a y-axis title.

We thank reviewer 2 for these suggestions. The two scattering curves are now both included in the graphs shown in Figure 1f and g as blue dashed lines, with the legend updated correspondingly. The y-axis title in figure 4c has now been included.

Reviewers' Comments:

Reviewer #1:

Remarks to the Author:

The new version of the manuscript is superior to the previous, and fully answers all of my questions. The study should proceed to publication.

Reviewer #2:

Remarks to the Author:

The authors have done a good job addressing all the reviewer comments and I believe the additional data and their changes to the text have strengthened the paper and further improved the clarity of an already very well written manuscript. The additional gel filtration analysis on dephosphorylated MLKL, mass spectrometry data pinpointing the phosphorylation sites, and the refined negative stain EM data further solidify the mechanistic link between MLKL phosphorylation and oligomerization into the functional tetrameric form. The paper represents a substantial advance in our understanding of the function of MLKL in necroptosis and is ready for acceptance in my opinion.